# `Terra`: A Multimodal Spatio-Temporal Dataset Spanning the Earth

**Wei Chen**[1]   **Xixuan Hao**[1]   **Yuankai Wu**[2]   **Yuxuan Liang**[1][*]
[1]The Hong Kong University of Science and Technology (Guangzhou)   [2]Sichuan University
onedeanxxx@gmail.com, yuxliang@outlook.com

## Abstract

Since the inception of our planet, the meteorological environment, as reflected through spatio-temporal data, has always been a fundamental factor influencing human life, socio-economic progress, and ecological conservation. A comprehensive exploration of this data is thus imperative to gain a deeper understanding and more accurate forecasting of these environmental shifts. Despite the success of deep learning techniques within the realm of spatio-temporal data and earth science, existing public datasets are beset with limitations in terms of spatial scale, temporal coverage, and reliance on limited time series data. These constraints hinder their optimal utilization in practical applications. To address these issues, we introduce `Terra`, a multimodal spatio-temporal dataset spanning the earth. This dataset encompasses hourly time series data from 6,480,000 grid areas worldwide over the past 45 years, while also incorporating multimodal spatial supplementary information including geo-images and explanatory text. Through a detailed data analysis and evaluation of existing deep learning models within earth sciences, utilizing our constructed dataset. we aim to provide valuable opportunities for enhancing future research in spatio-temporal data mining, thereby advancing towards more spatio-temporal general intelligence. Our source code and data can be accessed at https://github.com/CityMind-Lab/NeurIPS24-Terra.

## 1 Introduction

With the rapid development of remote sensing satellite systems [32, 27], radar monitoring devices [77, 65], and various advanced geographical observation technologies, spatio-temporal data [16, 79], particularly those pertaining to the Earth's environment and climate, are becoming increasingly available. Analyzing and mining valuable knowledge from such spatio-temporal data is crucial for many real-world applications, including environmental monitoring [37], disaster management [75], urban planning [40], and climate change assessment [76]. In the era of sensory artificial intelligence, an array of methods has been devised, ranging from conventional time series and spatial statistical analysis tools [80] to cutting-edge spatio-temporal deep learning models [30], for the analysis and utilization of domain-specific data. Despite remarkable achievements, there remains a huge gap between mainstream spatio-temporal data mining research [94] and the recent shift towards artificial general intelligence research [21], where the latter aims to address various challenges in a unified manner, while the generalizability and scalability challenge still lies in the former.

What causes this gap, or in other words, what measures can be taken to further advance research in spatio-temporal general intelligence? From historical experience, comprehensive, large-scale, high-quality datasets are crucial for the progress of any research community in any field. For instance, the ImageNet dataset [26], which has historically driven the development of the Computer Vision community, and the Common Crawl dataset [64], which is currently fostering growth in the

---

[*]Y. Liang is the corresponding author.

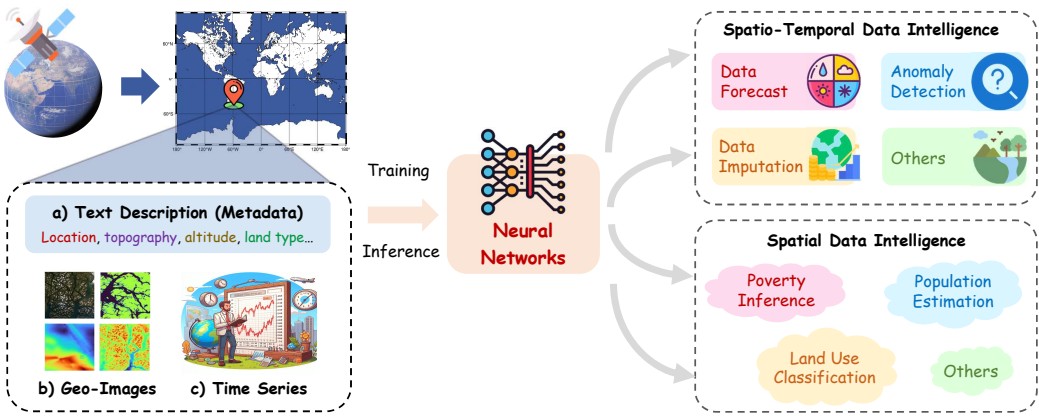

Figure 1: Overview of `Terra` and its application in spatio-temporal data intelligence.

Natural Language Processing community. The combination of these massive datasets with foundation model with billions parameters has led to remarkable breakthroughs in the generalization ability of vision and language models across various tasks [20, 63]. However, most of the work on spatio-temporal data mining focuses on developing advanced models [79], while neglecting the urgent need for comprehensive datasets themselves. Thus, there is an immediate necessity for a high-quality spatio-temporal dataset with sufficient breadth, granularity, capacity, and multimodal integration, allowing researchers to explore different methods at various scales and to move towards more general spatio-temporal intelligence.

To address these challenges, we introduce `Terra`, a public, large-scale, fine-grained, and multimodal dataset across spatio-temporal domains. The name is derived from the earth goddess in ancient Roman mythology. As shown in Figure 1, `Terra` rasterizes the Earth, integrates various data sources, and includes over 6820 billion hourly Earth meteorological observation time series data collected globally within raster grids from 1979 to 2024, as well as spatial multimodal geographic information supplements for all regions within global raster grids, including text descriptions and geographic images, aiming to advance spatio-temporal data analysis and spatial intelligence research. Specifically, `Terra` distinguishes itself significantly from previous spatio-temporal datasets with its comprehensive spatio-temporal coverage, outstanding quality, and diverse data modalities.

In summary, we condense its outstanding characteristics into three contributions: *(1) Large scale*, encompassing over 45 years of sequential observational information in terms of temporal range and global-level geographic information in terms of spatial range, ensuring the robustness, generalization, and reliability of models through large-scale data properties. *(2) Fine granularity*, supporting up to 3 hourly time granularity and up to 0.1° resolution spatial grid records, ensuring the practical feasibility of real-world application scenarios through fine-grained data properties. *(3) Multi Modality*, unlike previous datasets that only consisted solely of spatio-temporal observation records without exogenous data, we further provide a variety of multimodal supplementary data, including rich text and image data, enabling the exploration of large language and vision models in this field, thereby enhancing the interpretability of models.

## 2 Background

In this section, we review relevant works in spatio-temporal data mining, focusing on its challenges, various applications, and the limitations of existing datasets that support research in this field. It is worth noting that we particularly emphasize data types related to earth sciences. Other types of spatio-temporal data, such as vision-based video data [62], are beyond the scope of this paper.

### 2.1 Challenges and Applications of Spatio-Temporal Data Mining

Spatio-temporal data mining represents an interdisciplinary fusion of various fields such as spatio-temporal databases, machine learning, statistics, geography, meteorology, and information theory [31]. Specifically, spatio-temporal data refer to types of geographic entity data that exist at different scales

with spatio-temporal associations. These include spatial relationships, both metric (*e.g.*, distance) and non-metric (*e.g.*, topology, flow, and shape), temporal relationships (*e.g.*, before or after), and spatio-temporal relationships (*e.g.*, correlation and heterogeneity) that are explicitly or implicitly present in the data. In recent years, deep learning models such as recurrent neural networks [71], convolutional neural networks [93], and graph neural networks [40] achieve remarkable success in capturing the temporal and spatial dependencies in spatio-temporal data. These efforts lead to significant advances in various fields. For example, they have broad applications in environmental and climate areas (*e.g.*, precipitation forecasting [71] and air quality inference [53]), urban planning (*e.g.*, traffic flow prediction [40] and anomaly detection [36]), and human mobility (*e.g.*, travel recommendations [55] and personalized marketing [23]). However, applying deep models to spatio-temporal data is often more challenging. Firstly, spatio-temporal data are usually embedded in continuous space, different from the discrete space common in vision and language data. Secondly, spatio-temporal data often have high auto-correlation, contrasting with the traditional i.i.d. assumption of data samples. Lastly, spatio-temporal data have different scales of spatial and temporal resolution, and models trained on limited data often lack generalizability. Recently, with the success of foundational models [52, 18] (*e.g.*, large language models [11, 42] and diffusion models [67, 87]), researchers are exploring the construction of spatio-temporal foundational models to revolutionize this field. The aim is to achieve zero-shot inference and robust generalizability across different spatio-temporal tasks. For instance, in meteorological forecasting, Pangu [17] and GraphCast [48] provide unprecedented forecasting capabilities through pre-training on massive climate data. In urban computing [100], some studies [50, 51] combine the capabilities of large models to pioneer the development of foundational models for traffic. All these advancements enable researchers to uncover valuable insights into spatio-temporal patterns, continuously optimize the Earth's environmental systems, promote human economic and social development, and contribute to extensive interdisciplinary research.

## 2.2 Limitations of Existing Spatio-Temporal Datasets

Table 1 presents a comparison between the proposed `Terra` and other popular or latest spatio-temporal datasets. We next detail the improvements of `Terra` over others from five aspects.

- **Incomplete Analytical Perspectives:** Widely adopted datasets such as GeoLife [97], and more recent ones like GEO-Bench [47] and CityScape [34], have significantly contributed to spatial analysis research, including location recommendation and urban region analysis. At the same time, researchers have also performed various temporal analyses on time-series data like GluonTS [12], including tasks like prediction, imputation, and anomaly detection. However, these datasets are often only able to focus on singular spatial or temporal analyses. *In contrast, `Terra` offers comprehensive possibilities for spatio-temporal analysis from both perspectives.*

Table 1: Comparisons between `Terra` and other spatio-temporal datasets. Here, ✓ represents meeting a better standard, ✗ represents not meeting it, and ~ represents partially meeting or being convertible to meet it. Due to the difficulty of counting sizes from multiple data sources, we mark it with ◇.

| Dataset | Year | Accessibility | Volume | Large-Scale | | Fine-Granularity | | Multi-Modality | | |
|---|---|---|---|---|---|---|---|---|---|---|
| | | | | Spatial | Temporal | Spatial | Temporal | Time Series | Text | Image |
| Geolife [97] | 2010 | ✓ | 24M+ | ✗ | ✗ | ✓ | ✓ | ✓ | ✗ | ✗ |
| GluonTS [12] | 2020 | ✓ | 16M+ | ✗ | ~ | ✗ | ✓ | ✓ | ✗ | ✗ |
| NYC [88] | 2019 | ✓ | 22M+ | ✗ | ✗ | ✓ | ✓ | ✓ | ~ | ✗ |
| PEMS [72] | 2020 | ✓ | 42M+ | ✗ | ✗ | ✓ | ✓ | ✓ | ✗ | ✗ |
| SEVIR [77] | 2020 | ✓ | ◇ | ✗ | ✗ | ✓ | ✓ | ✗ | ✓ | ✓ |
| ML4Road [61] | 2024 | ✓ | 9M+ | ~ | ~ | ✓ | ✓ | ✓ | ✓ | ✗ |
| BioMassters [59] | 2024 | ✓ | 79M+ | ✗ | ~ | ✓ | ✗ | ✓ | ✗ | ✓ |
| ClimateSet [43] | 2024 | ✓ | ◇ | ✓ | ✓ | ~ | ~ | ✓ | ~ | ✗ |
| ClimSim [90] | 2024 | ✓ | 5.7B+ | ✓ | ✓ | ~ | ✓ | ✓ | ✗ | ✗ |
| Digital Typhoon [44] | 2024 | ✓ | 49B+ | ~ | ✓ | ✓ | ✓ | ~ | ✗ | ✓ |
| Mesogeos [46] | 2024 | ✓ | 1344B+ | ✗ | ✓ | ✓ | ✓ | ✓ | ~ | ~ |
| GEO-Bench [47] | 2024 | ✓ | ◇ | ✓ | ✗ | ✓ | ✗ | ✗ | ~ | ✓ |
| CityScape [34] | 2024 | ✗ | 65K+ | ✗ | ✗ | ✓ | ✗ | ✗ | ✓ | ✓ |
| ChatEarthNet [91] | 2024 | ✓ | ◇ | ✓ | ✗ | ✓ | ✗ | ✗ | ✓ | ✓ |
| `Terra` (ours) | 2024 | ✓ | 6820B+ | ✓ | ✓ | ✓ | ✓ | ✓ | ✓ | ✓ |

- **Restricted Access Opportunities:** Early spatial-temporal datasets often focused on human mobility (*e.g.*, Geolife [97] and NYC [88]) and intelligent transportation (*e.g.*, PEMS [72] and ML4Road [61]). However, these datasets typically involve privacy concerns and are held by a few companies or organizations with proprietary or restrictive access policies, resulting in restricted access to small datasets. Therefore, recent benchmarks has shifted towards spatio-temporal data in freely accessible fields such as Earth Sciences, leading to the emergence of numerous datasets such as BioMassters [59], ChatEarthNet [91], *etc. Similarly,* Terra *also benefits from this shift, allowing free access to these massive spatio-temporal data.*

- **Limited Spatio-Temporal Coverage:** Due to the labor and financial costs associated with data collection, existing datasets are often confined to specific cities or regions. Early examples, including NYC [88] and PEMS [72], generally cover only a few months of data for a single city or region. Recent earth science datasets, like SEVIR [77], Mesogeos [46] and Digital Typhoon [44], have significantly increased in scale but still struggle to achieve global coverage over several decades. This limitation results in insufficient geographic representation, impeding generalizability to other regions, and inadequate temporal representation, failing to capture seasonal or annual trends. *In contrast,* Terra *offers global spatial coverage and over 45 years of temporal coverage.*

- **Limited Spatio-Temporal Resolution:** Although some recent spatial-temporal datasets have reached considerable scales, they still fail to provide sufficiently granular spatial-temporal resolution. For instance, ClimateSet [43] only offers monthly climate records. These low sampling rates and resolutions further diminish their practical utility. *In contrast,* Terra *provides resolutions up to 0.1° spatially and 3-hour intervals temporally.*

- **Limited Multimodal Supplement:** Existing datasets often provide only basic spatial-temporal sequences, such as ClimSim [90], lacking rich multimodal supplementary information like text or images. This deficiency hinders comprehensive analysis and fails to meet the requirements for multimodal or advanced model design. *In contrast,* Terra *provides global-scale visual and textual information, serving as potential components for building foundational spatial-temporal models.*

## 3  Dataset Details

In this section, we formally introduce our proposed Terra dataset. As shown in Figure 2, our dataset consists of three modalities, each containing different types of data. We describe in detail the collection and processing methods of different modality data below to deepen the understanding of Terra. For more data introduction, analysis, statistics, visualization, and statement, see Appendix A.

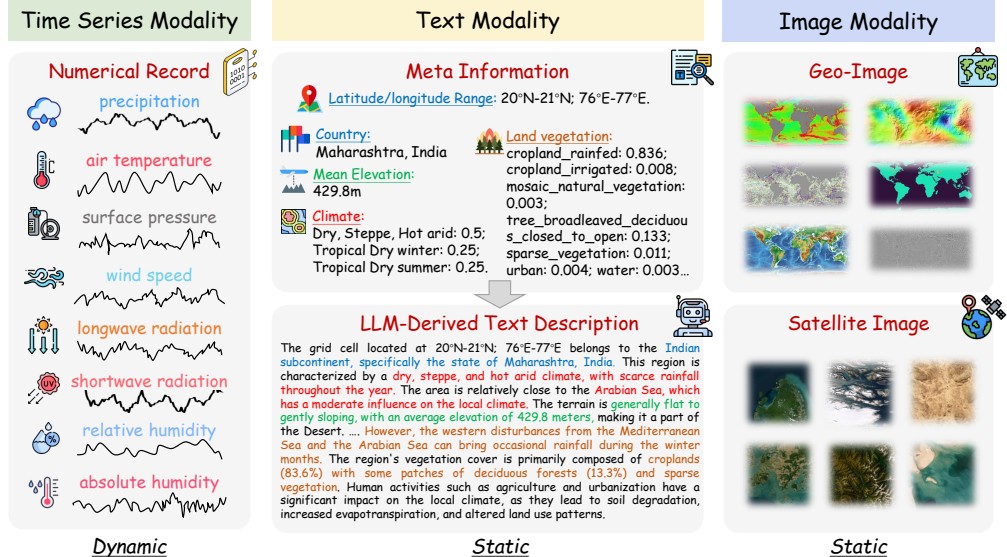

Figure 2: Different modality components of Terra. We provide the data with three temporal scales (3 hourly / daily / monthly), and three spatial scale (0.1° / 0.5° / 1°).

**Time Series Modality.** We obtain the time-series modality data for `Terra` from the *Global Water (GloH2O) Measurement Project* [5]. Specifically, we combine the past observation records from two products: Multi-Source Weather (MSWX) and Multi-Source Weighted-Ensemble Precipitation (MSWEP). MSWX is an operational high-resolution (3 hours, 0.1°), bias-corrected meteorological product, covering the global range from 1979 to 5 days before real-time. This product includes 10 meteorological variables: precipitation (mm/3h), air temperature (°C), daily minimum and maximum temperatures (°C), surface pressure (Pa), relative and specific humidity (% and g/g), wind speed (m/s), and

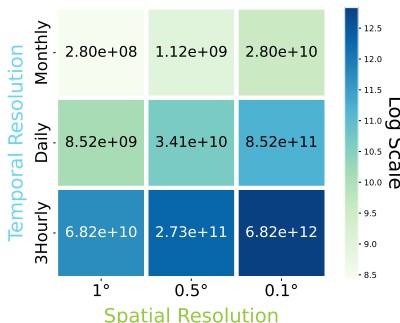

Figure 3: Dataset volume comparison.

downward shortwave and longwave radiation (W/m$^2$). As shown on the left side of Figure 2, we exclude the daily minimum and maximum temperatures due to their limited resolution. MSWEP, a global precipitation product, spans from 1979 to 3 hours before real-time with a resolution of 3 hours and 0.1°. Unlike MSWX, MSWEP uniquely combines gauge, satellite, and reanalysis data to provide the highest quality precipitation estimates at each location. Since MSWEP includes satellite data, its precipitation estimates may be more accurate than those of MSWX in regions with dense measurements and convection-dominated areas. Therefore, we replace MSWX precipitation records with MSWEP values. We select a time span covering 45 years from [01/01/1979 to 01/01/2024), equivalent to 540 months or 16,436 days. As a result, we obtain the largest dataset with a resolution of 0.1°, 3 hours, and a total of 6.82e+12 numerical records. Based on this, we further resample and combine the data at spatial resolutions of 0.5° and 1° and temporal resolutions of daily and monthly. This process yields a total of 9 variant, with the number of records for each dataset shown in Figure 3.

**Text Modality.** We first obtain the geographic text data within each raster region, mainly including climate, mean elevation, land vegetation, and the countries included. Specifically, We crawl the climate metadata from *Köppen climate classification project* [6], which reveal variations and changes of climate over the period 1901–2010. Given the slow pace of climate change, we utilize this data to represent current climate values. Climate types are represented by a two or three-letter combination, where the first letter denotes the major type (*e.g.*, tropical, dry, snow), and the second letter or third letter specifies subcategories (*e.g.*, fully humid, desert). Global elevation values are queried from *ETOPO2v2* [3], which combines topography, bathymetry, and shoreline data from both regional and global sources, enabling detailed, high-resolution renderings of the Earth's geophysical characteristics. We take the average of all data points within the indexed region, referred to as the mean elevation of the current indexed region. We also crawl Land Cover data from the *C3S Global Land Cover Product* [2], which classifies land cover into 38 categories (*e.g.*, cropland_rainfed and tree_broadleaved_deciduous_closed). we use data for 2022 year. For the land cover and climate type, we calculate the proportion within each region and provide it in percentage form. Regarding the country's affiliation, we referred to data from the world-geo-json repository [4]. Although these meta texts partially reflect the region's geographical characteristics, they lack comprehensive analysis and inference of potential spatial features (*e.g.*, how land cover types influence the area's climate and rainfall patterns). Recently, large language models (LLMs) have become essential for enhancing spatio-temporal data due to their integrated geographical knowledge, compressed through pre-training on extensive corpora [14, 42]. Consequently, we employ the state-of-the-art open-source language

---

*We have a series of* environment-related data *for a global grid divided into 1x1 degree latitude and longitude cells. Please generate a detailed text for given grid cell describing* **the main factors that may influence climate in that area**. *Include the following aspects:*

  ➢ **[Geographical Location]:** Which continent or country does it belong to?
  ➢ **[Climate Type]:** What type of climate does this area have (e.g., tropical, temperate, polar)?
  ➢ **[Ocean Influence]:** Is this area close to the ocean or large water bodies? How does the ocean influence the rainfall or weather?
  ➢ **[Terrain]:** What are the terrain features of this area (e.g., mountains, plains, deserts)?
  ➢ **[Monsoons]:** Is this area affected by monsoons? How do monsoons influence the weather?
  ➢ **[Airflows and Wind Belts]:** What are the main airflows and wind belts affecting this area?
  ➢ **[Vegetation Cover]:** What is the vegetation cover like in this area? How does vegetation influence weather?
  ➢ **[Human Activities]:** How do human activities (e.g., agriculture, urbanization) influence weather in this area?

*I will provide the* **latitude and longitude range** *of the area I want to describe, along with the* **vegetation type percentage, climate type and average altitude of the area**. *This information can be used as a reference to generate a more accurate description, but do not just focus on these points. If there is no climate type, it means this area is primarily ocean. Please write the description in a paragraph, and avoid saying other things.*

Please generate a text description for <latitude range> and <longitude range> in <country> according to the above structure.
  √ **Landvegetation:** ***. √ **Climate:** ***. √ **Mean Elevation:** ***.

Figure 4: A example of spatial prompt engineering.

model LLaMA3 [58] to generate supplementary textual information. Given the open challenge of hallucinations in LLMs [38], resulting inaccuracies can introduce noise into downstream tasks. To mitigate this problem, we designed a spatial prompt engineering, as shown in Figure 4. This technique suggests querying factual meta text-related to country, climate, land vegetation, and mean elevation characteristics as auxiliary prompts. This approach aims to provide LLMs with comprehensive information, facilitating more accurate extraction of surface environmental data. Figure 5 presents statistics and visual insights of the selected LLM text modality data. We also discuss the necessity of our prompting technique and compare different LLM choices in Appendix A.3.

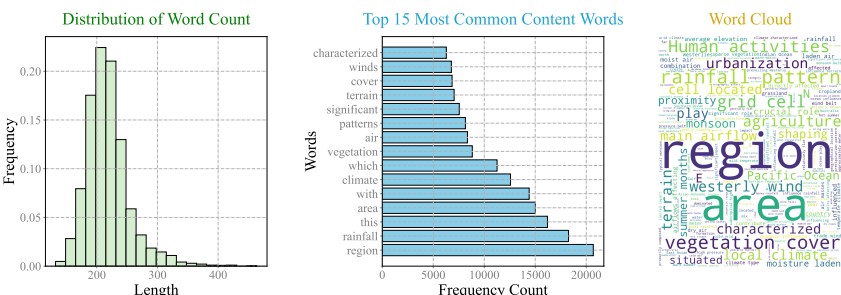

Figure 5: Statistical and visual insights of text modality data.

**Image Modality.** We further adopt the Mercator projection [10] to map the Earth into grids at different spatial resolutions and crawl relevant geographic image information for each grid cell. Specifically, we first access geographic datasets provided by remote machines of GMT [9] and utilize the PyGMT [7] toolkit to draw geo-images according to specified longitude and latitude regions. In particular, we select common geographic images including Earth Geoid (coinciding with the mean sea level and extending into the interior of continents), Earth Free-Air Anomaly Errors (normalizing observed values and height correction, converting gravity values to gravity anomalies referenced to the same latitude geoid), Earth Magnetic Anomaly (obtained by subtracting the global magnetic field from the Earth's core main magnetic field and its induced magnetic field after subtracting the variable magnetic field, resulting in lithospheric magnetic field), Earth Mask (referring to surface water-land geographical features), Earth Relief (containing observed topography and terrain inferred through height gravity), Earth Vertical Gravity Gradient (referring to the vertical derivative of gravity for detecting geological structures and positions of small geological bodies). All these converted geo-images are commonly used for exploring Earth sciences and summarizing regional geographic information. Additionally, satellite remote sensing images are usually another excellent visual descriptive imagery of geographic information, apart from the aforementioned types, and can also be obtained for each grid under corresponding image information using ArcGIS [1].

**Discussion.** Despite our best efforts to obtain rich spatio-temporal data from multiple sources, several unavoidable issues still persist: Firstly, our image and text modalities still do not support a spatial resolution of 0.1° due to the enormous time and monetary costs associated with this scale. Secondly, as we utilize LLM models to generate text data, the inevitable obsolescence of the latest LLM capabilities exists. Finally, the acquired satellite remote sensing images may have outdated and unstable redistribution restrictions. For the first issue, we hope to continually invest time and monetary resources to highlight higher-resolution text and image modal data in future versions. Regarding the second issue, we first empirically studied the suitability of existing generated text (see Appendix A.3) and look forward to regularly update and utilize the latest and best open-source LLM to generate new text data, selectively replacing existing text. For the third issue, we also suggest exploring alternative satellite image products from other open-source communities, such as Sentinel-2 [8], for updates.

**Potential Applications Summary.** We summarize a range of potential application scenarios for our proposed `Terra` dataset, encompassing but not limited to those enumerated in the Table 2. Spanning remote sensing, urban indicator prediction, time series forecasting, and beyond, the diverse modalities and extensive volume of the `Terra` dataset present limitless application possibilities. We aspire for the community to leverage `Terra` to foster significant advancements in spatial-temporal data mining.

**Data License.** The `Terra` dataset is made available under the CC BY-NC 4.0 International License: https://creativecommons.org/licenses/by-nc/4.0. Our code and dataset are released under the MIT License: https://opensource.org/licenses/MIT. The license of any baselines and subdata sources used in this paper should also be verified on their official repositories.

Table 2: Potential application scenarios for `Terra` dataset. All single modalities can be associated with geo-coordinates.

| | Modality | Application Examples | Method Examples |
|---|---|---|---|
| **Single Modality** | (Spatial-) Time Series | Time Series Forecasting, Imputation, Classification[1], Spatio-Temporal Forecasting, Interpolation[2], ... | Moirai[1] [81], TimesNet[1] [82], UrbanDiT[2] [15] |
| | Image | Remote Sensing Representation Learning[1], Location Embedding[2], Geo-Localization[3], Super-Resolution for Remote Sensing, ... | Cross-Scale MAE[1] [74], Scale-MAE[1] [66], G3[1] [39], SatCLIP[2] [45], GeoCLIP[3] [78], CSP[2] [56] |
| | Text | Geo-Language Model[1], Geo-Text Mining[2], ... | GeoLLM[1] [57], K2[1] [25], [2] [33, 68, 13] |
| **Multi Modality** | (Spatial-) Time Series + Image | Image-enhanced Spatial[1] / Temporal[2] Prediction, ... | VisionTS[2] [22], [1] [73, 49, 19] |
| | (Spatial-) Time Series + Text | LLM-enhanced Spatio-Temporal Forecasting, ... | UrbanGPT [50], Time-LLM [41], Promptcast [85] |
| | Image + Text | Urban Region Profiling[1], Remote Sensing LLM[2], Satellite Image-Text Retrieval[3], ... | UrbanCLIP[1] [86], UrbanVLP[1] [34], EarthGPT[2] [95], UrbanCross[3] [98] |
| | (Spatial-) Time Series + Image + Text | World Model[1], Urban Plan[2], Urban Foundation Model[3], ... | UGI[1] [84], CityGPT[2] [29], UFM[3] [96] |

# 4 Use Cases

To further demonstrate the practicality and versatility of our dataset, we have selected its use in two key tasks: spatio-temporal analysis and spatial analysis. These tasks are classical examples in spatio-temporal data mining, with the former providing insights into joint modeling of spatial and temporal, and the latter supporting common applications in spatial modeling. Through these simple application cases, our goal is to illustrate the versatility and usability of `Terra`.

## 4.1 Spatio-Temporal Analysis Task

**Problem Definition.** Each record in a grid is a multivariate time series $x \in \mathbb{R}^{T \times C}$, capturing dynamic observations of $C$ measurement features over $T$ time steps. Here, $N$ regions with spatially correlated locations constitute a spati-otemporal tensor $X \in \mathbb{R}^{N \times T \times C}$. Spatio-temporal forecasting predicts signals $X \in \mathbb{R}^{N \times T^f \times C}$ for $T^f$ future time steps across $N$ variables, utilizing $T^h$ steps historical time series $X \in \mathbb{R}^{N \times T^h \times C}$ (and an optional spatial correlation graph $\mathcal{G}$ among recorded regions).

**Setup.** Here, we choose precipitation prediction as a representative example. We extract spatio-temporal precipitation sequences from time-series modality data with a temporal resolution of 1 day and spatial resolution of 1°, covering a temporal span of 26 years and global spatial extent, forming the *Global* dataset. Subsequently, we construct five smaller country datasets representing five continents by selecting representative countries: the United Kingdom (*UK*), the United States (*USA*), China (*CN*), South Africa (*SA*), and Australia (*AUS*). For all datasets, we partition the dataset into training, validation and test sets as 6:2:2. Then, we select four categories of popular methods as baselines, including time-series models (**TimesNet** [82], **FEDformer** [99], **PatchTST** [60], **DLinear** [92]), spatio-temporal prediction models (**STAEformer** [54], **STID** [70], **GWNet** [83], **STGCN** [89]), precipitation-specific prediction model **ConvLSTM** [71], and historical mean method **HI** [24]. For different methods, in order to adapt to the task, we make appropriate feature, structure and hyperparameter adjustments to achieve the best results. We conduct three prediction scenarios: predicting precipitation for the next 7, 15, and 30 days based on the historical 30-day precipitation sequences, using mean absolute error (MAE) and root mean square error (RMSE) to evaluate prediction performance. All experiments are conducted three times, and the mean values are reported. For more detailed information about the experimental setup, please refer to Appendix A.

**Result Analysis.** Table 3 presents the MAE and RMSE test results for specific horizons of 7, 15, and 30 days, along with the average performance across all prediction horizons. The simple method **HI** performs the worst as it completely ignores temporal dependency and spatial correlation, relying solely on the last lagged value from historical records. Additionally, the state-of-the-art **TimesNet** model in time-series prediction and advanced spatio-temporal prediction model **STID** achieve the best and second-best performance, respectively. This could be attributed to their incorporation of embedding information of time and

Figure 6: Global performance comparison.

Table 3: Spatio-Temporal Forecasting Performance. **Red**: the best, Blue: the second best.

| Methods | | TimesNet | | FEDformer | | PatchTST | | DLinear | | STAEformer | | STID | | GWNet | | STGCN | | ConvLSTM | | HI | |
|---|---|---|---|---|---|---|---|---|---|---|---|---|---|---|---|---|---|---|---|---|---|
| Metric | | MAE | RMSE | MAE | RMSE | MAE | RMSE | MAE | RMSE | MAE | RMSE | MAE | RMSE | MAE | RMSE | MAE | RMSE | MAE | RMSE | MAE | RMSE |
| UK | 7 | 2.812 | 4.681 | 3.350 | 4.618 | 3.286 | 4.616 | 3.313 | 4.566 | 3.224 | 4.559 | 3.232 | 4.579 | 3.219 | 4.592 | 3.229 | 4.577 | 3.250 | 4.600 | 4.023 | 5.466 |
| | 15 | 2.821 | 4.697 | 3.372 | 4.695 | 3.327 | 4.673 | 3.324 | 4.611 | 3.241 | 4.554 | 3.233 | 4.560 | 3.235 | 4.668 | 3.249 | 4.599 | 3.285 | 4.661 | 4.100 | 5.547 |
| | 30 | 2.817 | 4.693 | 3.438 | 4.706 | 3.346 | 4.702 | 3.328 | 4.603 | 3.226 | 4.638 | 3.226 | 4.519 | 3.235 | 4.620 | 3.260 | 4.655 | 3.285 | 4.652 | 4.067 | 5.532 |
| | Avg | 2.816 | 4.690 | 3.376 | 4.659 | 3.316 | 4.666 | 3.320 | 4.591 | 3.234 | 4.581 | 3.232 | 4.553 | 3.229 | 4.629 | 3.245 | 4.627 | 3.266 | 4.645 | 4.026 | 5.479 |
| USA | 7 | 1.356 | 4.578 | 3.319 | 7.171 | 3.296 | 7.262 | 3.263 | 7.296 | 3.167 | 7.138 | 3.158 | 7.021 | 3.179 | 7.005 | 3.181 | 7.041 | 3.209 | 7.254 | 4.146 | 8.300 |
| | 15 | 1.370 | 4.609 | 3.342 | 7.205 | 3.320 | 7.326 | 3.280 | 7.380 | 3.178 | 7.128 | 3.165 | 7.071 | 3.188 | 7.039 | 3.198 | 7.077 | 3.230 | 7.290 | 4.122 | 8.232 |
| | 30 | 1.400 | 4.675 | 3.413 | 7.329 | 3.383 | 7.495 | 3.336 | 7.498 | 3.210 | 7.224 | 3.189 | 7.142 | 3.219 | 7.134 | 3.237 | 7.218 | 3.267 | 7.405 | 4.145 | 8.263 |
| | Avg | 1.371 | 4.613 | 3.348 | 7.212 | 3.321 | 7.327 | 3.282 | 7.364 | 3.178 | 7.147 | 3.165 | 7.057 | 3.184 | 7.077 | 3.197 | 7.088 | 3.225 | 7.315 | 4.102 | 8.175 |
| CN | 7 | 3.147 | 6.689 | 4.721 | 7.903 | 4.654 | 8.021 | 4.679 | 8.002 | 4.547 | 7.833 | 4.538 | 7.822 | 4.553 | 7.865 | 4.558 | 7.854 | 4.576 | 7.883 | 5.945 | 8.970 |
| | 15 | 3.155 | 6.679 | 4.730 | 7.911 | 4.664 | 8.038 | 4.682 | 7.991 | 4.548 | 7.833 | 4.538 | 7.814 | 4.550 | 7.880 | 4.570 | 7.816 | 4.577 | 7.877 | 5.977 | 9.005 |
| | 30 | 3.163 | 6.674 | 4.751 | 7.880 | 4.668 | 8.044 | 4.679 | 7.988 | 4.537 | 7.847 | 4.535 | 7.804 | 4.552 | 7.845 | 4.565 | 7.862 | 4.579 | 7.876 | 5.975 | 9.018 |
| | Avg | 3.152 | 6.677 | 4.723 | 7.890 | 4.654 | 8.017 | 4.672 | 7.981 | 4.544 | 7.827 | 4.536 | 7.806 | 4.544 | 7.839 | 4.555 | 7.825 | 4.566 | 7.856 | 5.888 | 8.888 |
| SA | 7 | 2.067 | 3.455 | 2.350 | 3.400 | 2.335 | 3.424 | 2.317 | 3.394 | 2.297 | 3.394 | 2.319 | 3.340 | 2.281 | 3.358 | 2.278 | 3.409 | 2.300 | 3.408 | 3.084 | 4.194 |
| | 15 | 2.075 | 3.459 | 2.366 | 3.411 | 2.343 | 3.443 | 2.323 | 3.404 | 2.302 | 3.355 | 2.289 | 3.372 | 2.284 | 3.354 | 2.283 | 3.402 | 2.308 | 3.395 | 3.074 | 4.175 |
| | 30 | 2.093 | 3.476 | 2.392 | 3.444 | 2.362 | 3.466 | 2.333 | 3.421 | 2.291 | 3.384 | 2.310 | 3.351 | 2.299 | 3.354 | 2.285 | 3.417 | 2.300 | 3.417 | 3.095 | 4.199 |
| | Avg | 2.076 | 3.460 | 2.365 | 3.412 | 2.343 | 3.441 | 2.322 | 3.402 | 2.295 | 3.362 | 2.308 | 3.351 | 2.294 | 3.346 | 2.282 | 3.408 | 2.303 | 3.404 | 3.077 | 4.182 |
| AUS | 7 | 1.830 | 3.160 | 2.171 | 3.184 | 2.157 | 3.206 | 2.131 | 3.208 | 2.109 | 3.230 | 2.109 | 3.161 | 2.101 | 3.208 | 2.107 | 3.188 | 2.104 | 3.221 | 2.790 | 3.900 |
| | 15 | 1.838 | 3.164 | 2.170 | 3.194 | 2.164 | 3.214 | 2.136 | 3.212 | 2.111 | 3.214 | 2.102 | 3.151 | 2.102 | 3.207 | 2.113 | 3.184 | 2.112 | 3.230 | 2.790 | 3.898 |
| | 30 | 1.844 | 3.173 | 2.184 | 3.227 | 2.172 | 3.241 | 2.145 | 3.232 | 2.118 | 3.227 | 2.100 | 3.177 | 2.102 | 3.196 | 2.115 | 3.190 | 2.120 | 3.242 | 2.806 | 3.920 |
| | Avg | 1.835 | 3.162 | 2.171 | 3.191 | 2.160 | 3.215 | 2.135 | 3.211 | 2.109 | 3.225 | 2.098 | 3.157 | 2.098 | 3.201 | 2.107 | 3.188 | 2.110 | 3.213 | 2.768 | 3.876 |
| 1st Count | | 30 | | 0 | | 0 | | 0 | | 2 | | 6 | | 2 | | 0 | | 0 | | 0 | |

date, which intuitively aids in predicting precipitation fluctuations, which tend to be substantial. An interesting and surprising observation is that spatio-temporal prediction models do not outperform time-series prediction models, even the **ConvLSTM** model specifically designed for precipitation prediction. Typically, this contradicts intuition as methods considering dynamic spatial topology features are expected to outperform time-series prediction methods where variables are independent, as observed in literature [69]. However, in our precipitation prediction experiment, it is evident that most spatio-temporal and time-series prediction models yield similar results, failing to adequately capture spatio-temporal trends. One possible reason could be that precipitation is often a non-stationary time series with extreme fluctuations, making it challenging to capture clear patterns. Hence, the community needs to explore how to incorporate more external information to aid prediction and increase interpretability. As a solution, our multimodal dataset `Terra` provide a potential avenue for further investigation. Figure 6 further illustrates the global prediction results, where a similar phenomenon is observed. The only difference is that spatio-temporal models such as **STAEformer**, **GWNet**, and **STGCN** fail to perform effectively due to their high spatial memory consumption, further motivating research into more efficient spatio-temporal models in the future.

## 4.2 Spatial Analysis Task

### 4.2.1 Location based Spatial Variable Prediction

**Problem Definition.** Spatial variables refer to indicators related to geospatial coordinates, such as precipitation, wind speed, and population. Using current location embedding technology, we can predict environmental indicators solely based on geospatial coordinates and pre-learned location data, which is crucial for spatial analysis. Formally, location-based spatial prediction aims to predict $L$ spatial indicators of $N$ locations, denoted as $Y \in \mathbb{R}^{N \times L}$, given the global latitude and longitude coordinates $X \in \mathbb{R}^{N \times 2}$ of $N$ locations.

**Setup.** Here we use *precipitation* (mm/day), *wind speed* (m/s), and *air temperature* (°C) as representative indicators to conduct spatial variable prediction experiments. Specifically, we select global scale and 1° spatial resolution data (Aggregation by 0.1°), and use the daily average value from 2020 to 2022 as the ground-truth indicator value. We use three representative position embedding models: **SatCLIP** [45], **GeoCLIP** [78], and **CSP** [56]. The position encoder is frozen during training, and only *Multilayer Perceptron* is added on top of it for linear probing [35]. In addition, the dataset samples are randomly shuffled and divided into training, validation, and test sets in a ratio of 7:1:2. The performance indicators used to compare them are mean square error and mean absolute error.

**Result Analysis.** Table 4 presents a comparison of different models across three representative environmental spatial variables. As we can see, **SatCLIP** demonstrates superior performance, attributable to its satellite pretraining dataset. We also display an intuitive visual comparison in Figure 7. In contrast to **GeoCLIP**, which utilizes geo-tagged street-view images for pretraining,

Table 4: Spatial Prediction Performance. **Red**: the best, Blue: the second best.

| Methods | Precipitation | | Wind Speed | | Temperature | |
|---|---|---|---|---|---|---|
| | MSE | MAE | MSE | MAE | MSE | MAE |
| CSP | 0.997 | 0.775 | 0.965 | 0.787 | 0.909 | 0.761 |
| GeoCLIP | 0.162 | 0.249 | 0.212 | 0.326 | 0.021 | 0.095 |
| SatCLIP | 0.010 | 0.052 | 0.036 | 0.116 | 0.002 | 0.024 |

**SatCLIP** could be offered more pertinent semantic information for downstream environment-related tasks. Conversely, **CSP** focuses on pretraining location encoders for specific applications, which diminishes its adaptability to new downstream tasks. Overall, these results corroborate the suitability and practical utility of our `Terra` dataset for environment-related spatial variable prediction, thereby reinforcing its potential for advancing research in spatial-temporal data mining.

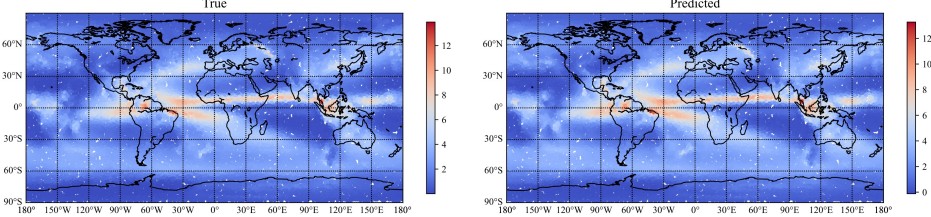

Figure 7: SatCLIP predicted global-scale precipitation.

### 4.2.2 Vision-Language based Spatial Variable Prediction

**Problem Definition.** To demonstrate the multimodal nature of the `Terra`, we follow the recent paradigm of spatial variable prediction based on vision-language pre-training [34, 86], which provides a comprehensive geographic vision of a region through satellite images, and provides an overview of the region's inherent knowledge through text descriptions, thereby enhancing spatial variable prediction. Formally, vision-language based spatial prediction aims to predict $L$ spatial indicators of $N$ locations, denoted as $Y \in \mathbb{R}^{N \times L}$, given visual and textual pairs $X \in \mathbb{R}^{N \times \langle I, T \rangle}$ of $N$ locations.

**Setup.** We follow the same experimental setup as in Sec. 4.2.1. However, in the context of vision-language pre-training, making predictions on a global scale poses significant challenges in terms of computational resources. Thus, similar to Sec. 4.1, we select three representative countries(*USA*, *CN*, *AUS*) for experiments. We select two representative VLP models in the spatio-temporal domain, **UrbanVLP** [34] and **UrbanCLIP** [86], as well as the classic general baseline **CLIP** [63]. Similarly, the dataset is split into training, validation, and test sets in a 7:1:2 ratio, and the performance metrics used for comparison are: coefficient of determination ($R^2$) and mean squared error (MSE).

Table 5: Results of Vision-Language based Spatial Variable Prediction. UrbanVLP* dentoes that we leverage UrbanVLP without its street-view branch. **Red**: the best, Blue: the second best.

| Datasets | Precipitation | | | | | | Wind Speed | | | | | | Temperature | | | | | |
|---|---|---|---|---|---|---|---|---|---|---|---|---|---|---|---|---|---|---|
| Methods | CN | | USA | | AUS | | CN | | USA | | AUS | | CN | | USA | | AUS | |
| Metric | $R^2$ | MSE | $R^2$ | MSE | $R^2$ | MSE | $R^2$ | MSE | $R^2$ | MSE | $R^2$ | MSE | $R^2$ | MSE | $R^2$ | MSE | $R^2$ | MSE |
| **CLIP** | 0.483 | 0.537 | 0.359 | 0.631 | 0.321 | 0.683 | 0.587 | 0.429 | 0.519 | 0.483 | 0.266 | 0.712 | 0.513 | 0.454 | 0.575 | 0.422 | 0.178 | 0.992 |
| **UrbanCLIP** | 0.617 | 0.418 | 0.409 | 0.577 | 0.383 | 0.623 | 0.674 | 0.352 | 0.579 | 0.425 | 0.340 | 0.657 | 0.685 | 0.343 | 0.650 | 0.344 | 0.210 | 0.981 |
| **UrbanVLP*** | **0.745** | **0.279** | **0.589** | **0.402** | **0.680** | **0.323** | **0.774** | **0.244** | **0.750** | **0.252** | **0.591** | **0.407** | **0.791** | **0.228** | **0.802** | **0.195** | **0.352** | **0.804** |

**Result Analysis:** Table 5 illustrates the performance of three spatial variable predictions on three countries. Overall, these models demonstrated similar performance to those presented in [34]. The distinct performance of different models effectively highlights the consistency of our dataset. Specifically, the performance trends for each metric vary across the three countries. For example, the USA exhibits relatively poor performance in precipitation prediction, possibly due to its status as the country with the most diverse climate types in the world, which affects precipitation patterns. Conversely, Australia's suboptimal performance in temperature prediction may be attributed to its unusual geographic situation, being surrounded by oceans while having an inland desert climate. Additionally, due to being trained on data from a limited number of countries, the performance is slightly inferior to location-based models, which use pretrained encoders on global-scale datasets.

## 5 Conclusions

This work introduces the `Terra`, a multimodal spatio-temporal dataset. `Terra` is a comprehensive dataset encompassing various meteorological data spanning the earth, covering 6,480,000 grid regions over the past 45 years. It includes spatio-temporal observations along with multimodal spatial information such as geo-images and explanatory texts. Based on a thorough introduction of the data and analysis of experimental results, we highlight the significant impact of the `Terra` dataset on advancing spatio-temporal data mining research and its potential for progressing towards spatio-temporal general intelligence.

## Acknowledgments and Disclosure of Funding

The authors would like to thank the anonymous reviewers for their valuable comments. This work is mainly supported by the National Natural Science Foundation of China (No. 62402414). This work is also supported by the Guangzhou-HKUST(GZ) Joint Funding Program (No. 2024A03J0620), Guangzhou Municipal Science and Technology Project (No. 2023A03J0011), the Guangzhou Industrial Information and Intelligent Key Laboratory Project (No. 2024A03J0628), and a grant from State Key Laboratory of Resources and Environmental Information System, and Guangdong Provincial Key Lab of Integrated Communication, Sensing and Computation for Ubiquitous Internet of Things (No. 2023B1212010007).

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

# SUPPLEMENTARY MATERIAL
## TERRA: A MULTIMODAL SPATIO-TEMPORAL DATASET SPANNING THE EARTH

## TABLE OF CONTENTS

# A   More Dataset Details

We provide a comprehensive supplementary introduction to the `Terra` dataset in this section, covering data composition, statistics, visualization and analysis. Moreover, We also discuss the risks existing in the text descriptions derived from LLM and the rationality of our solutions. Lastly, We conclude by presenting statements on data availability and access links.

## A.1   Data Composition and Statistics

After completing the entire process of acquiring and cleaning time series data from different sources, we further resample the data to obtain time series data of different temporal and spatial resolutions. For each spatial unit area at different spatial resolutions, we also supplement them with other meta text elements, generated text knowledge, geo-images, and remote sensing images in a multimodal context. This section provides a detailed description of the various data types contained in each spatial unit of the dataset, their structure, and key attributes, ensuring a comprehensive understanding of the content of the dataset and its potential applications.

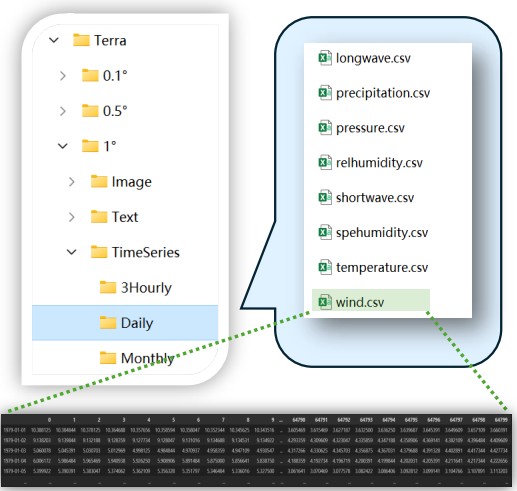

Figure 8: Example of folder structure and file contents for time series modality.

**Time Series.** Our dataset is structured based on spatial resolution, with initial directories containing data at 1° / 0.5° / 0.1° resolutions. The 1° and 0.5° resolutions are obtained by downsampling 0.1° data by a factor of 10 and 5, respectively. For the time series data within each spatial resolution directory, we further categorize them into subdirectories based on temporal resolution, including monthly / daily / 3 hourly data. The monthly and daily data are obtained by downsampling the 3 hourly data by a factor of 8 and 8 times the number of days in a month (28-31 days). Specifically, under each temporal resolution, there are usually 8 sub-files, each recording climate index values from 00:00 on January 1, 1979, to 00:00 on January 1, 2024 (inclusive start, exclusive end), spanning 45 years. These files contain data on precipitation, air temperature, surface pressure, relative and specific humidity, wind speed, and downward shortwave and longwave radiation. Note that for individual sub-files that take up a large amount of storage, we will split them into multiple files based on the year. The complete folder directory and the contents of a single file are shown in Figure 8, where each file under the subdirectory is stored in .csv format, and the shape is summarized in Table 6.

Table 6: Shape of a single file at different spatial and temporal resolutions.

| Temporal \ Spatial | Spatial Resolution | | |
|---|---|---|---|
| | 0.1° | 0.5° | 1° |
| 3 Hourly | [131488, 6480000] | [131488, 259200] | [131488, 64800] |
| Daily | [16436, 6480000] | [16436, 259200] | [16436, 64800] |
| Monthly | [540, 6480000] | [540, 259200] | [540, 64800] |

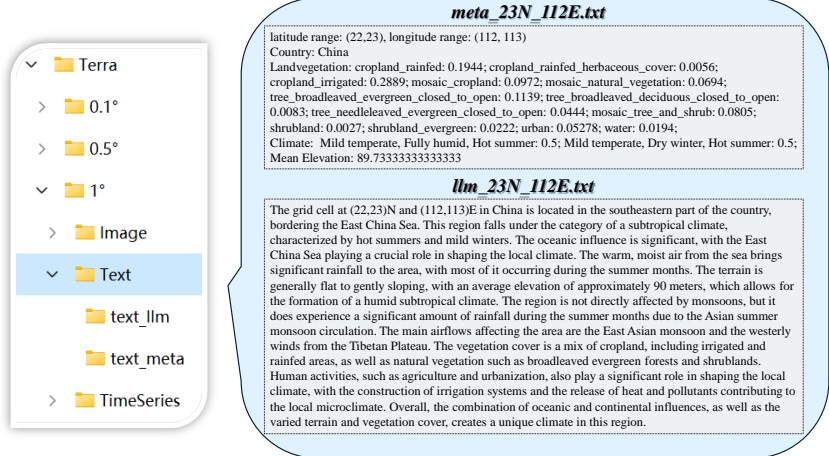

Figure 9: Example of folder structure and file contents for text modality.

**Text.** As mentioned in the previous section, our text dataset also includes initial directories at different spatial resolutions. Due to the significant time and cost involved, we currently only provide text modal information at 1° and 0.5° spatial resolutions. For the text data at each spatial resolution, we provide two types of text description files. The first type of description file is meta-text data. We crawl the corresponding spatial unit area metadata for the project as introduced in the main body, including latitude and longitude ranges, countries, land types, climate conditions, and average elevation, organized and stored together in a *.txt* file. The file name is named after the upper left

Table 7: Processing methods and raw information of the meta text.

| Meta Name | Processing | Raw Information | Area |
|---|---|---|---|
| Range | Split Segment | [−90 ∼ +90] S / N,
[−180 ∼ +180] W / E | Global |
| Country | Index Search | Afghanistan, Antarctica, Andorra
...,
Yemen, Zimbabwe, Zambia | Global |
| Mean Elevation | Calculate Average | [−8775.47 ∼ +5371.13] meter | Global |
| Land Vegetation | Proportional Allocation | no_data, cropland_rainfed, cropland_rainfed_herbaceous_cover,
cropland_rainfed_tree_or_shrub_cover, cropland_irrigated,
mosaic_cropland, mosaic_natural_vegetation,
tree_broadleaved_evergreen_closed_to_open,
tree_broadleaved_deciduous_closed_to_open,
tree_broadleaved_deciduous_closed,
tree_broadleaved_deciduous_open,
tree_needleleaved_evergreen_closed_to_open,
tree_needleleaved_evergreen_closed,
tree_needleleaved_evergreen_open,
tree_needleleaved_deciduous_closed_to_open,
tree_needleleaved_deciduous_closed,
tree_needleleaved_deciduous_open,
tree_mixed, mosaic_tree_and_shrub,
mosaic_herbaceous, shrubland, shrubland_evergreen,
shrubland_deciduous, grassland, lichens_and_mosses,
sparse_vegetation, sparse_tree, sparse_shrub,
sparse_herbaceous, tree_cover_flooded_fresh_or_brakish_water,
tree_cover_flooded_saline_water,
shrub_or_herbaceous_cover_flooded,
urban, bare_areas, bare_areas_consolidated,
bare_areas_unconsolidated, water, snow_and_ice | Global |
| Climate | Proportional Allocation | Tropical rain forest, Tropical monsoons,
Tropical savanna with dry summer,
Tropical savanna with dry winter,
Desert (arid), Steppe (semi-arid),
Mild temperate with dry summer,
Mild temperate with dry winter,
Mild temperate, fully humid,
Snow with dry summer, Snow with dry winter,
Snow, fully humid, Tundra, Frost,
Hot arid, Cold arid, Hot summer,
Warm summer, Cool summer, Cold summer | Mainland |

corner latitude and longitude of the unit area (for example, the file for the 22-23N, 112-113E grid area is named *meta_23N_112E.txt*). The specific processing methods and original information for all categories in the meta text are shown in Table 7. The second type of description file is LLM-derived text description. Similarly, for each unit spatial area, we use the spatial cue engineering technology introduced in the text and generate climate description information for each grid area using the latest open-source LLM model LLama3, storing the content in a *.txt* file (the file for the same area grid is named *llm_23N_112E.txt*). The complete folder directory and the contents of a single text modality file are shown in Figure 9.

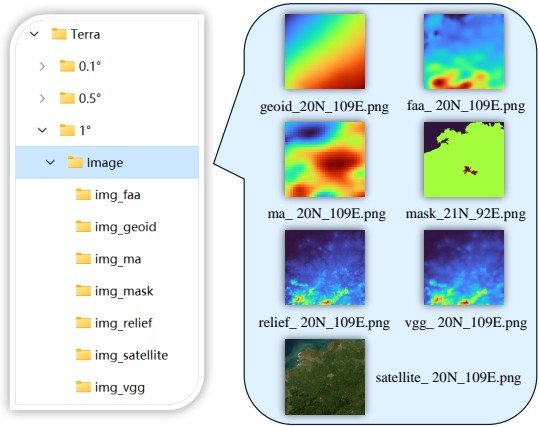

Figure 10: Example of folder structure and file contents for image modality.

**Image.** Similarly, our image dataset is also contained in initial directories with different spatial resolutions. Due to significant time and cost constraints, we currently provide modal information images at 1° and 0.5° spatial resolutions. The image data provides descriptions of both overall and detailed aspects of all spatial regions. For each spatial resolution, we offer various types of image information. Specifically, as described in the text, there are seven main categories of images, namely the geoid area map, spatial error area map, lithosphere magnetic field area map, hydrological area map, topography area map, and gravity geological area map, along with the satellite remote sensing area map. The first six can be considered as geo-image categories, primarily describing inherent properties of the Earth's terrain, topography, land, and sea. The last one is classified as a remote sensing image category, mainly used to describe overall regional geographical information. Different types of image folders are named sequentially as (*img_geoid*, *img_faa*, *img_ma*, *img_mask*, *img_relief*, *img_vgg*, *img_satellite*), and for each type, individual spatial region images are named based on the upper left corner's longitude and latitude (for example, the relief map for the 19-20N, 109-110E grid is named *relief_20N_109E.png*). The complete folder directory and the contents of a single image modality file are shown in Figure 10. Due to the adoption of the Mercator projection, distortions are unavoidable for images in the polar regions; therefore, we disregard high latitude areas in the Arctic and Antarctic circles. Additionally, for the *img_mask* files, completely oceanic and completely terrestrial regions cannot be projected as image types, so they are omitted. The summary of all file types, areas, and levels is presented in Table 8.

Table 8: Basic information about different image.

| Image Name | Type | Area | Level | Source time |
|---|---|---|---|---|
| img_geoid | geo-image | $[-80 \sim +80]$ S / N, $[-180 \sim +180]$ W / E | Code: 01m | 2008 |
| img_faa | geo-image | $[-80 \sim +80]$ S / N, $[-180 \sim +180]$ W / E | Code: 01m | 2019 |
| img_ma | geo-image | $[-80 \sim +80]$ S / N, $[-180 \sim +180]$ W / E | Code: 02m | 1946-2014 |
| img_mask | geo-image | $[-80 \sim +80]$ S / N, $[-180 \sim +180]$ W / E | Code: 15s | - |
| img_relief | geo-image | $[-80 \sim +80]$ S / N, $[-180 \sim +180]$ W / E | Code: 01s | 2019 |
| img_vgg | geo-image | $[-80 \sim +80]$ S / N, $[-180 \sim +180]$ W / E | Code: 01m | 2019 |
| img_satellite | satellite-image | $[-85 \sim +85]$ S / N, $[-180 \sim +180]$ W / E | Zoom: 10 / 11 | 2023-2024 |

## A.2 Data Visualization and Analysis

We further selected representative regions used in the experiment and visualized some modal data metadata corresponding to them, aiming to provide local intuitive insights.

**Time Series.** As shown in Figure 11, we randomly selected precipitation time series data from three spatial regions and aligned them for visualization. The first column displays the precipitation variations in these three spatial regions over more than 20 years. Among them, Grid 98 and Grid 99 are geographically adjacent, while Grid 3 is geographically distant from the other two grids. It can be observed that the precipitation amounts in Grids 98 and 99 exhibit similar patterns of change, reflecting spatial correlation. In contrast, Grid 3 shows a distinctly different precipitation pattern compared to the other two grids, indicating spatial heterogeneity. Further Seasonal and Trend decomposition using Loess (STL) of the precipitation time series data for each spatial region reveals similar observations in terms of trend, seasonality, and residuals.

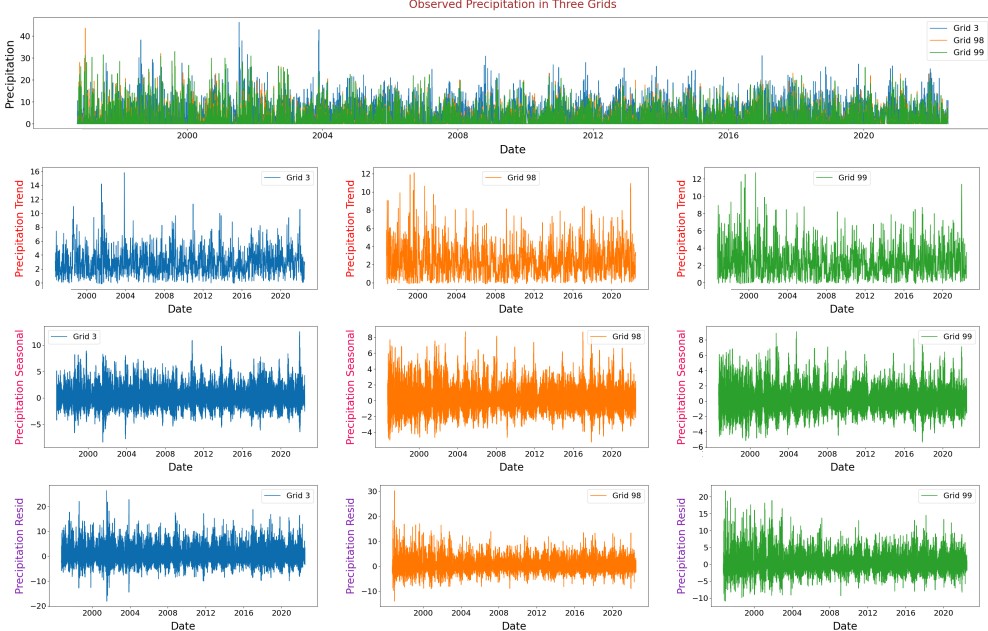

Figure 11: Statistical and visual insights of time series modality data.

**Text.** Figure 5 presents statistics and visual insights of the selected LLM text modality data. As we can observe, the text length predominantly centers around 200, which is longer than that found in current state-of-the-art datasets such as ChatEarthNet dataset [91], whose highest frequency distribution occurs at 155 (for texts generated by GPT-3.5). Our most common content words are primarily related to geography, environment, and climate, reflecting the consistency between the text content and the thematic focus of the `Terra` dataset.

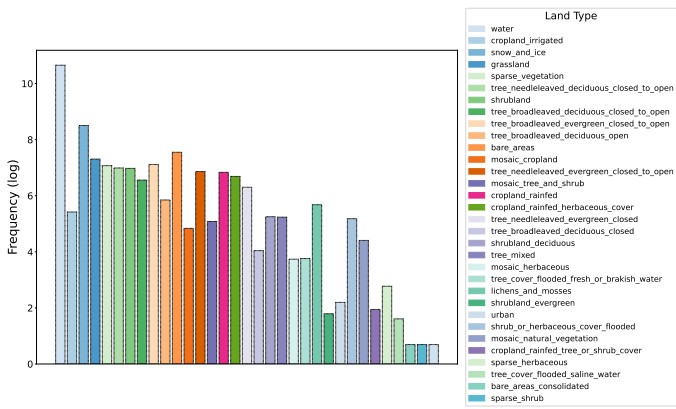

Figure 12: Land type frequency diagram.

**Image.** Due to the difficulty of direct image visualization, we use metadata on land types and climate types as proxy labels for spatial area images. Specifically, we select the predominant land or climate type in the current spatial area as the true label. As shown in Figure 12, we first visualize global surface types. It is well known that oceans cover approximately 71% of the Earth's surface area, which aligns with our visualization finding that water has the highest frequency. As illustrated in Figure 13, we further visualize the climate type proportions for all regions. It is evident that polar frost and snow climates, as well as snow, fully humid and cool summer have the highest proportions. In contrast, temperate climates with dry summers and cool summers, and temperate climates with dry winters and cool summers, have the lowest proportions.

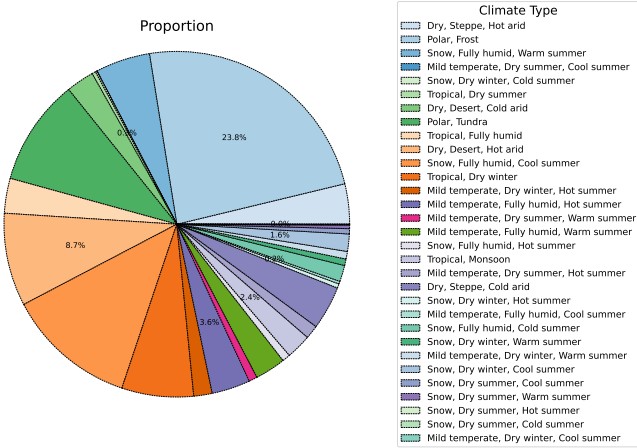

Figure 13: Climate type proportion diagram.

## A.3 Discussion of LLM-Derived Text Description

In this section, we further discuss the technology of LLM-derived text description. Specifically, we first analyze the spatial prompt engineering strategy and rationality we designed, and then compare the impact of different open source LLMs on the quality of text descriptions.

**The Suitability of Current Text Data.** The text generation methods of LLMs are promising; however, it remains unclear to what extent the text generated by LLMs is reliable, even with the use of metadata. One suggested approach for quantification is to extract text embeddings from the generated text, using either the LLM itself or BERT, and then train a multi-class classifier or regressor on these embeddings to predict each metadata attribute present in the txt metadata file. The accuracy of the classifier serves as a rough proxy for the illusion rate of each attribute (the frequency at which the LLM omits or incorrectly alters attributes). We refer to paper [28] to calculate the illusion rate of the generated text. Due to the diversity of metadata, we simultaneously train both a regressor and a classifier. For the regression task, we predict the elevation and latitude/longitude coordinates of the current area. For the classification task, we identify the dominant land vegetation type in the region. Additionally, we choose BERT as the backbone encoder. The dataset is split in a 7:1:2 ratio. Table 9 presents the NRMSE and accuracy metrics for several text regions used in this study. The experimental results indicate that our spatial prompting engineering approach achieves approximately 70% retention and success rates, effectively mitigating the illusion problem to some extent.

Table 9: The Suitability study of experimental text data

| Country | NRMSE | Accuracy |
|---------|-------|----------|
| USA | 0.0519 | 0.764 |
| AUS | 0.056 | 0.724 |
| SA | 0.0705 | 0.696 |

**The necessity of Spatial Prompt Engineering.** As shown in Figure 14, here we provide an illustration example of comparison of generated text with prompt with meta data and without meta data. As observed, the absence of metadata can cause large language models (LLMs) to experience hallucinations, resulting in factual inaccuracies, generalized climate descriptions, erroneous terrain depictions, and incomplete vegetation information. Conversely, the provision of precise metadata can effectively steer LLMs towards generating accurate and detailed descriptions.

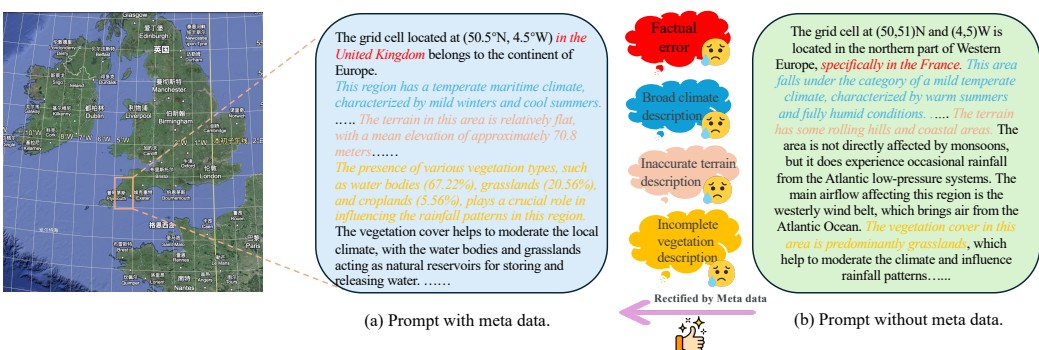

Figure 14: Comparison of generated text with prompt with meta data and without meta data.

**The impact of different LLM choices.** We compare four common popular large language models, including the open-source language models LLaMA3-8B, Vicuna-13b, Gemini-1.5 Pro, and the closed-source language model GPT3.5, as shown in a simple comparison example in Figure 15. Among them, LLaMA3-8B and GPT3.5 have achieved similarly excellent results. Compared to other open-source large language models, LLaMA3-8B can generate more accurate descriptions of climate types and provide more detailed subdivisions of various components, which has been confirmed by numerical data. In addition, the acceptable performance of other models also proves the effectiveness of our prompts, highlighting the potential of using LLM to extract and elucidate compressed geographical spatial knowledge.

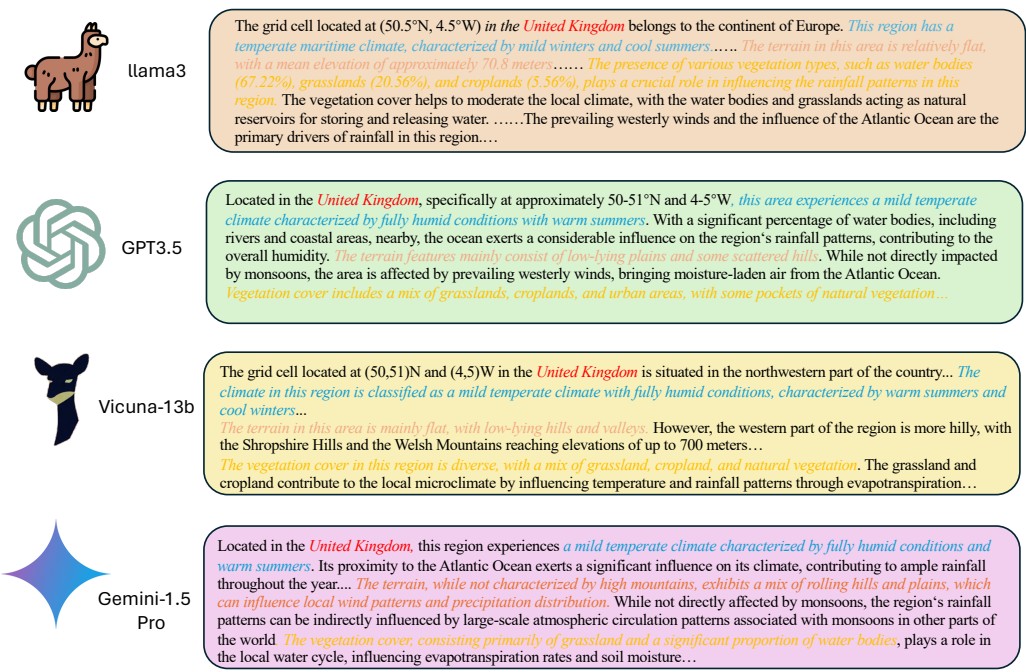

Figure 15: Comparison of generated text with different LLMs.

### A.4 Data Statements and Accountability.

The code and data used for the experiment can be accessed in the repository `https://github.com/CityMind-Lab/NeurIPS24-Terra`. The official dataset will be hosted on the Hugging Face repository `https://huggingface.co/datasets/onedean/Terra`. Our code and dataset follow the CC BY-NC 4.0 International License. All authors confirm the data license and commit that the dataset will only be used for academic research.

## B  More Experimental Details

### B.1  Training Resources & Code Implementation.

Our running environment consists of a Linux server equipped with a $2\times$ AMD EPYC 7763 128-Core Processor CPU (512GB memory) and $8\times$ NVIDIA RTX A6000 (48GB memory) GPUs. To carry out benchmark testing experiments, all baselines are set to run for a duration of 24 hours by default, with specific timings contingent upon the method.

### B.2  Spatio-Temporal Analysis Experiment

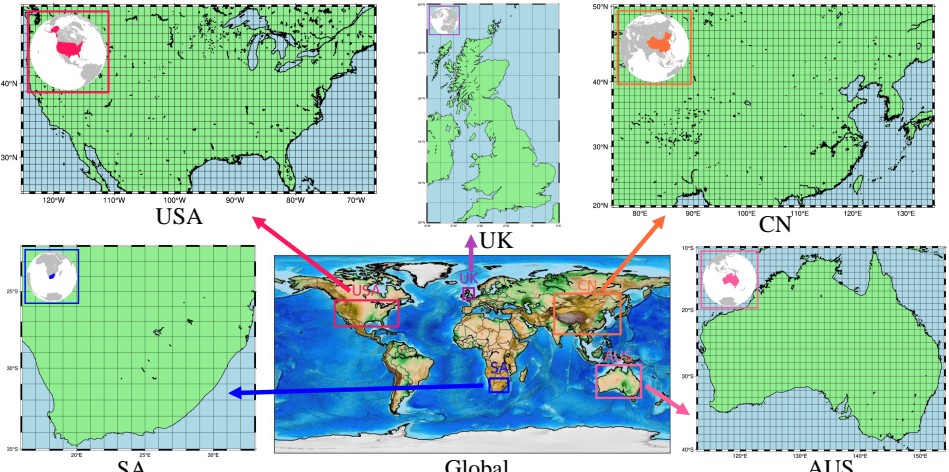

Figure 16: Visualization of dataset regions.

**Detail Datasets.** As shown in Table 10, we provide the statistical information of the datasets used in this experiment. Five representative countries were selected from the five continents around the world. However, since we directly used rectangles to cover the selected countries, the area covered may exceed the actual country area, or miss areas outside the main continent (such as Alaska State in the USA). Below are the specific ranges of different countries and basic geographical information introductions. To further illustrate, we give an intuitive visualization as shown in Figure 16.

**Detail Baselines.** The essence of spatio-temporal forecasting lies in predicting the time series of multiple spatial regions, thus the available method choices include time series forecasting and spatio-temporal forecasting methods. In addition, we also include models specifically designed for precipitation prediction and the traditional HI method. Below is a brief introduction to each method:

- TimesNet [82]: This paper presents a novel time series forecasting model that consistently achieves state-of-the-art performance in various time series analysis tasks by capturing the multi-periodicity and complex variations of time series through the transformation of one-dimensional time series into two-dimensional tensors.

- FEDformer [99]: The paper introduces a Transformer model that combines seasonal trend decomposition and frequency enhancement for long-term time series forecasting, improving the efficiency and accuracy of predictions by applying the Transformer in the frequency domain.

Table 10: Spatio-temporal analysis dataset statistics. *Please note that due to the use of rectangular selection, the current latitude and longitude coverage only includes parts of the country's territory, and may also include some territories and oceans of other countries. The dataset name is used for identification purposes only and does not represent actual national territories.*

| Dataset | Spatial Coverage | | Temporal Coverage | Resolution | | Variable |
|---------|------------------|------------------|-------------------|---------|----------|----------|
|         | Bottom-left | Top-right |                   | Spatial | Temporal |          |
| Global | -90°S, -180°W | 90°N, 180°E | 1996/10/1-2022/6/30 | 1° | Daily | Precipitation |
| UK | 50°N, -8°W | 60°N, 2°E | 1996/10/1-2022/6/30 | 1° | Daily | Precipitation |
| USA | 25°N, -125°W | 49°N, -67°W | 1996/10/1-2022/6/30 | 1° | Daily | Precipitation |
| CN | 20°N, 75°E | 50°N, 135°E | 1996/10/1-2022/6/30 | 1° | Daily | Precipitation |
| SA | -35°S, 16°E | -22°S, 33°E | 1996/10/1-2022/6/30 | 1° | Daily | Precipitation |
| AUS | -40°S, 113°E | -10°S, 154°E | 1996/10/1-2022/6/30 | 1° | Daily | Precipitation |

- PatchTST [60]: A highly efficient Transformer-based model for multivariate time series forecasting and self-supervised representation learning is proposed. This model significantly enhances the accuracy of long-term predictions by segmenting the time series into small pieces and processing each channel independently.

- DLinear [92]: The paper introduces a simple linear model called DLinear, which decomposes the time series into trend and remainder sequences and uses a two-layer linear network for direct multi-step forecasting, often outperforming existing complex Transformer models.

- STAEformer [54]: A new type of spatio-temporal Transformer model is introduced, which significantly enhances the accuracy of spatio-temporal forecasting by incorporating spatio-temporal adaptive embeddings. The STAEformer achieves state-of-the-art performance on six real-world datasets.

- STID [70]: This paper presents a simple and effective baseline model for multivariate spatio-temporal forecasting, which addresses the indistinguishability of samples in spatial and temporal dimensions by adding spatial and temporal identity information. Based on a simple multi-layer perceptron (MLP), it achieves optimal performance and efficiency.

- GWNet [83]: This paper introduces a novel graph neural network architecture called Graph WaveNet for spatial-temporal graph modeling. It effectively captures hidden spatial dependencies and long sequence time trends in the data through adaptive dependency matrices and stacked dilated one-dimensional convolutional components.

- STGCN [89]: A new deep learning framework for spatio-temporal forecasting, STGCN, is introduced. It combines graph convolution and gated temporal convolution to capture spatio-temporal correlations in the network, achieving excellent performance on multiple real-world traffic datasets.

- ConvLSTM [71]: This paper proposes a new convolutional LSTM network for precipitation forecasting. By extending the traditional fully connected LSTM to include convolutional structures for input-to-state and state-to-state transitions, ConvLSTM is better equipped to capture correlations in spatio-temporal sequences. https://github.com/jhhuang96/ConvLSTM-PyTorch

- HI [24]: This paper introduces historical inertia (HI) as a new baseline for long-term time series forecasting. Research indicates that even when HI is used directly as the output, it can achieve improvements on four public real-world datasets.

We have chosen the benchmark toolbox BasicTS+ https://github.com/zezhishao/BasicTS/tree/master designed specifically for spatio-temporal and time series forecasting as our code framework. However, it should be noted that TimesNet, STAEformer, and STID require date embeddings. Since we are predicting the future 30 days based on the past 30 days, we use "day of month" and "month of year" as substitutes for the original more fine-grained time embeddings to capture trends on a daily and monthly basis. Additionally, for GWNet and STGCN, spatial graphs are constructed by connecting each region with its neighboring regions. Finally, following the original paper on ConvLSTM, we implement it in the code framework for ease of comparison. For all methods, we adhere to the original default parameter settings and make appropriate adjustments for optimal performance.

**Detail Metrics.** Our evaluation is conducted on re-normalized data, employing metrics such as MAE, MSE. Formally, assuming $n$ represents the indices of all observed samples, $y_i$ denotes the $i$-th actual sample, and $\hat{y}_i$ is the corresponding prediction, these metrics are formulated as following:

$$\text{MAE} = \frac{1}{n}\sum_{i=1}^{n}|y_i - \hat{y}_i|, \quad \text{RMSE} = \sqrt{\frac{1}{n}\sum_{i=1}^{n}(y_i - \hat{y}_i)^2}$$

## B.3 Spatial Analysis Experiment

**Detail Datasets.** As shown in Table 11, we summarize the statistical information of the datasets used for spatial analysis tasks. In contrast to spatio-temporal analysis tasks, we utilize daily averages with a finer spatial resolution of 0.1° from 2020 to 2022 to represent spatial indicators. Specifically, we selected three indicator variables: precipitation, wind speed, and air temperature.

Table 11: Spatial analysis dataset statistics. *Please note that due to the use of rectangular selection, the current latitude and longitude coverage only includes parts of the country's territory, and may also include some territories and oceans of other countries. The dataset name is used for identification purposes only and does not represent actual national territories.*

| Dataset | Spatial Coverage | | Temporal Coverage | Resolution | | #LLM Text | #Satellite Image | Variable |
|---|---|---|---|---|---|---|---|---|
| | Bottom-left | Top-right | | Spatial | Temporal | | | |
| Global | -90°S, -180°W | 90°N, 180°E | 2020/1/1-2022/12/31 | 0.1° → 1° | Daily-Mean | / | / | Precipitation, Wind Speed, Air Temperature |
| USA | 25°N, -125°W | 49°N, -67°W | 2020/1/1-2022/12/31 | 1° | Daily-Mean | 1,392 | 1,392 | Precipitation, Wind Speed, Air Temperature |
| CN | 20°N, 75°E | 50°N, 135°E | 2020/1/1-2022/12/31 | 1° | Daily-Mean | 1,800 | 1,800 | Precipitation, Wind Speed, Air Temperature |
| AUS | -40°S, 113°E | -10°S, 154°E | 2020/1/1-2022/12/31 | 1° | Daily-Mean | 1,230 | 1,230 | Precipitation, Wind Speed, Air Temperature |

**Detail Baselines.** For spatial analysis tasks, we have chosen two types of spatial variable prediction methods based on location and visual-language. A brief overview of all methods is as follows:

- CSP [56], also known as Contrastive Spatial Pre-Training, is a robust and innovative multi-modal, self-supervised pre-training methodology, capitalizing on extensive collections of un-labeled geo-tagged imagery. This approach significantly advances the acquisition of location-specific representations, thereby facilitating their application in few-shot learning scenarios. https://github.com/gengchenmai/csp

- GeoCLIP [78] pioneers the worldwide geo-localization task through an image-to-GPS retrieval methodology, explicitly aligning image features with their corresponding GPS locations. By integrating positional encoding with random Fourier features, its location encoder effectively encodes GPS coordinates, thus reducing spectral bias in multi-layer perceptrons (MLPs). https://github.com/VicenteVivan/geo-clip

- SatCLIP [45] introduces the first global-coverage, general-purpose pretrained geographic location encoder, utilizing Satellite Contrastive Location Image Pretraining. SatCLIP distills spatially varying visual patterns from globally distributed satellite data into an implicit neural representation within a compact and efficient neural network, achieving excellent performance across a wide range of downstream tasks. https://github.com/microsoft/satclip

- CLIP [63], as a milestone in the field of Vision-Language Pretraining (VLP), it has demonstrated that cross-modal learning with web-scale data can exhibit outstanding performance. CLIP excels in various domains such as image-text retrieval, image-text generation, and possesses zero-shot capabilities. https://github.com/openai/CLIP

- UrbanCLIP [86] is the pioneering framework that integrates textual modality knowledge into urban region analysis. It demonstrates that the comprehensive textual data generated from Large Multimodal Models is an siginificant supervision signal to urban area representations. Through a contrastive learning-based encoder-decoder architecture, UrbanCLIP injects textual knowledge into visual representations by incorporating contrastive loss as well as language modeling loss, making it a versatile and robust approach for urban region profiling. https://github.com/StupidBuluchacha/UrbanCLIP

- UrbanVLP [34] introduces a pioneering framework for urban region representation learning that comprehensively explores multi-granularity cross-modal alignment. This framework also facilitates automatic text generation and calibration through the application of PerceptionScores, thereby ensuring the high quality of generated texts. This innovative approach has exhibited exceptional performance across six downstream tasks, spanning environmental, social, and economic domains.

For all methods, we followed the default parameters and architectural designs as stated in the original papers, and conducted experiments using their official codes. The only modification made was to UrbanVLP, as this method requires street view images which were not effectively obtainable in our dataset. Therefore, the encoder branch used for street view images was removed while the rest remained unchanged.

**Detail Metrics.** To assess the spaital prediction performance, we also adopt three commonly used evaluation metrics: mean squared error (MSE), mean absolute error (MAE) and coefficient of determination ($R^2$). MAE and MSE are calculated in a similar way as in the previous section, and both of these metrics are better when they are smaller. The $R^2$ is a statistical measure that indicates how well a regression model fits the data. It represents the proportion of the variance in the dependent variable that is predictable from the independent variables. The formulation for $R^2$ is:

$$R^2 = 1 - \frac{\sum_{i=1}^{n}(y_i - \hat{y}_i)^2}{\sum_{i=1}^{n}(y_i - \bar{y})^2}$$

where $\bar{y}$ is the mean of the actual values. An $R^2$ value closer to 1 indicates that the model explains a large proportion of the variance in the data, meaning the model has a good fit. The advantage of $R^2$ is that it provides a measure of how well the model explains the variability of the data. However, it can be less useful for non-linear relationships and is sensitive to outliers. In summary, MSE, MAE, and $R^2$ have their respective strengths and weaknesses. MSE and MAE primarily measure the magnitude of prediction errors, while $R^2$ evaluates the explanatory power of the model.

## C More Discussion

**Potential for Higher Resolution.** Currently, the time series modality in our dataset has achieved a spatial resolution of $0.1°$, while the corresponding text and image modalities have not reached this level. Specifically, the main challenge in achieving higher spatial resolution for text lies in the reliability and redundancy of the Large Language Model generating the text, as text descriptions are mainly related to the climate, terrain, and environment of the current area. Fine-grained spatially adjacent regions may have a large amount of redundant text descriptions, posing greater challenges for noise removal. On the other hand, the main adjustment in achieving higher spatial resolution for images lies in the significant time and monetary costs required to handle and manage such fine-grained data. However, increasing the resolution to a spatial scale of $0.1°$ will exponentially increase the amount of data, necessitating advanced storage solutions, more powerful computing resources, and substantial financial investment. To address this issue, we are committed to continuously investing time and resources in gradually improving the resolution of various modalities in our dataset. This includes leveraging advancements in data storage technologies, optimizing data processing algorithms, and exploring cloud platforms that can support scalable storage.

**Risk of LLM Obsolescence.** The use of LLMs to generate explanatory text data is a double-edged sword. While LLMs provide a powerful tool for creating comprehensive and detailed textual data, they are also prone to rapid obsolescence due to the fast-paced advancements in AI research and technology. As newer, more advanced LLMs emerge, the text generated by older models may become outdated or less accurate, potentially compromising the quality of the dataset. Moreover, keeping the text data relevant and accurate necessitates regular updates and replacements, which can be resource-intensive. To mitigate these risks, we hope to regularly update our dataset with the latest and best open-source LLMs. This involves a systematic review process to identify advancements in LLM capabilities, followed by the selective replacement of existing text data. By doing so, we ensure that our dataset remains at the forefront of technological innovation and continues to provide high-quality, reliable data for research and practical applications.

**Instability of Satellite Distribution.** Due to potential redistribution constraints, the reliance on commercial satellite remote sensing imagery introduces a certain degree of instability in data acquisition. These constraints may stem from changes in commercial policies, geopolitical factors,

or fluctuations in the satellite data market dynamics. For example, Esri grants recipients of Esri information contained in the esri.com website the right to freely copy, redistribute, rebroadcast, and/or retransmit this information for personal non-commercial purposes (including educational, classroom use, academic and/or research purposes) [2]; however, these permissions may expire or encounter issues over time. Therefore, in future iterations, we will explore alternative satellite image products from the open-source community, such as Sentinel-2 provided by the European Space Agency (ESA). Open-source satellite data offers a more stable, cost-effective solution, ensuring continued access to high-quality imagery reliably. Furthermore, establishing partnerships with academic and governmental institutions that provide open access to satellite data can further enhance the robustness and stability of our dataset.

**Timeliness of Spatial Text and Images.** One potential assumption of our dataset is that spatial information is stable and not time-dependent. However, despite spatial information typically being static or slowly changing, it can undergo significant changes over time. For instance, rapid urbanization can lead to substantial changes in land use, infrastructure, and population density. Areas that were once rural may urbanize, altering the spatial features captured in early images. Furthermore, natural disasters such as earthquakes, floods, and hurricanes can completely transform landscapes, necessitating image updates to accurately reflect these changes. These examples underscore the importance of ensuring the timeliness of spatial images and related textual data in a dataset. Nevertheless, obtaining dynamic temporal text and images is currently impractical, not only due to the challenge of accessing valid sources but also because of the massive storage and crawling costs involved. As a future research direction, we look forward to developing automated systems for detecting changes in spatial data and triggering updates of images and text. Machine learning algorithms can be utilized to identify areas undergoing significant changes, prompting a review and update of the corresponding data.

# D   Broader Impact

The introduction of this `Terra` dataset will undoubtedly have a profound impact in the fields of earth science and deep learning, and these impacts will be reflected in many aspects.

**Positive Impact**

- **More Accurate Weather Forecasting and Environmental Monitoring:** The Terra dataset provides hourly time series data spanning 45 years globally, enabling meteorologists and environmental scientists to predict weather patterns, climate trends, and the likelihood of natural disasters more accurately. This is crucial for preparedness, ecological protection, and ensuring human safety.

- **Enhanced Training and Application of Deep Learning Models:** The scale and multimodal nature of this dataset will provide rich data resources for the development of deep learning models. Training and evaluating on spatio-temporal data can improve existing models and lay a solid foundation for future spatio-temporal general intelligence.

- **Promotion of Scientific Research and Interdisciplinary Collaboration:** The open sharing of the Terra dataset will facilitate collaboration between fields such as Earth science, computer science, and data science. Researchers can jointly utilize this dataset to explore the complexity of the Earth system, advancing interdisciplinary research.

**Negative Impact**

- **Data Bias and Misinterpretation:** In constructing and utilizing the Terra dataset, there may be issues of data bias and misinterpretation. For example, uneven distribution of observation stations or sensor malfunctions may result in inaccurate or incomplete data for certain regions, leading to misinterpretation of environmental characteristics.

- **Technological Dependency and Data Dominance:** Over-reliance on the Terra dataset may lead to neglect of other data sources, weakening diversity and reliability. When using this dataset, it is important not to overly rely on it but to complement it with other data resources for comprehensive analysis and validation.

---

[2]https://www.esri.com/en-us/legal/copyright-proprietary-rights

