# OpenReview forum: "Terra: A Multimodal Spatio-Temporal Dataset Spanning the Earth"
_NeurIPS.cc/2024/Datasets_and_Benchmarks_Track — NeurIPS 2024 Track Datasets and Benchmarks Poster_

### Official Review · Reviewer_uuE1 · 2024-07-20
**Terra: A Multimodal Spatio-Temporal Dataset Spanning the Earth**

**Rating:** 7
**Confidence:** 3
**Correctness:** Need to strengthen
**Clarity:** Needs to improve

**Review:**

The pros and cons of this paper:
- The paper has provided a case study to illustrate the proposed approach.
- The paper illustrated the data set details.
- The paper listed the limitations of existing spatio-temporal datasets.

Cons:
-  The paper shall strengthen the technical details of proposed approach.
- The paper lacks a discussion on the limitation of this paper.

**Strengths:**

- The paper has provided a case study to illustrate the proposed approach.
- The paper illustrated the data set details.
- The paper listed the limitations of existing spatio-temporal datasets.

**Additional Feedback:**

NA

**Documentation:**

Yes

**Limitations:**

No. Shall add it.

**Opportunities For Improvement:**

-  The paper shall strengthen the technical details of proposed approach.
- The paper lacks a discussion on the limitation of this paper.

**Relation To Prior Work:**

Yes

**Summary And Contributions:**

In this paper, the authors introduced Terra, a multimodal spatio-temporal dataset spanning the earth. This dataset encompasses hourly time
series data from 6,480,000 grid areas worldwide over the past 45 years, while also incorporating multimodal spatial supplementary information including geo-images and explanatory text. Terra will help the community to gain a deeper understanding and more accurate
forecasting of environmental shifts.

The Terra system has three contributions: (1) Large scale; (2) Fine granularity; (3) multi-modality.

---

> ### Author Rebuttal · Authors · 2024-08-15
>
> Dear Reviewer uuE1:
>
> We really appreciate your hard work during the review period. We are also pleased that you agree with the thoroughness and richness of our work. Here, we address your concerns in detail:
>
> + **W1**: The paper lacks a discussion on the limitations of this paper.
>
> >   A gentle reminder is that you seem to have missed our appendix materials, which can address many of your concerns. As per official requirements, we included it in the supplementary section, which can be found in the **Supplementary Material**. We apologize for any unnecessary confusion caused by the current document division. Nevertheless, we will briefly summarize and respond to the discussion on limitations here to help you quickly understand:
> >>  **Potential for Higher Resolution**: Although we have made our best effort to collect the largest, highest-resolution, and most diverse data sources possible, the benefits of even higher resolution in practical applications are still worth considering. That being said, this is almost unrealistic at present due to storage and computing costs. We believe the current version is sufficient for researchers to conduct extensive research and use.
>
> >> **Risk of LLM Obsolescence**: Although we used the most advanced open-source language model, LLaMA3, before the deadline, the progress of large language models is so rapid that there is even a more advanced LLaMA3.1 version recently. However, this is inevitable. We addressed this concern by designing prompt templates and the hallucination evaluation section mentioned by Reviewer 1. Additionally, we plan to update the use of the best language models regularly to achieve better text quality.
>
> >> **Instability of Data Distribution**: Considering that some of our images may encounter unstable distribution restrictions, we discussed alternative options for using more replaceable data sources.
>
> >> **Timeliness of Spatial Information**: One of our basic assumptions is that spatial information is relatively stable in the long term. However, spatial information will inevitably change due to economic development or natural disasters. Currently, adding dynamic spatial information is impractical in terms of storage and access costs, with limited benefits. We also suggest developing algorithms for dynamic detection in the future to help improve this.
>
> >  Note that all of these limitations are almost unavoidable, and we have done our best with the resources available to us. Additionally, for each limitation, we have discussed potential solutions for the future. We believe that if the community shows interest in this work, it will collaboratively advance and maintain the project, making it more popular and user-friendly.
>
> + **W2**: The paper shall strengthen the technical details of the proposed approach.
>
> >  We discussed in detail our dataset construction methods, components, and analysis in the appendix (see **Supplementary Material**). We also covered the data, methods, techniques, and evaluation tools used in our spatio-temporal and spatial analysis experiments. Considering the limited space of the main text, we are sorry that some technical details have to be left in the appendix. We hope you can find the complete answers in the appendix, and we welcome any further questions you might have.
>
> Thank you once again for your invaluable suggestions to improve our work!

---

> > ### Author Response · Authors · 2024-08-28
> > **End of rebuttal is coming soon (Aug. 31). Sincere Gratitude from the Authors**
> >
> > Dear Reviewer uuE1,
> >
> > Since the end of the rebuttal is coming very soon (Aug. 31), we would like to inquire if our response addresses your primary concerns. If it does, we kindly request you to reconsider the confidence or score. If you have any additional suggestions, we are more than willing to engage in further discussions and make necessary improvements to the paper.
> >
> > Thanks again for dedicating your time to enhancing our paper!

---

### Official Review · Reviewer_fRfp · 2024-07-20
**Great contribution. Missing how the dataset may compare to other ones.**

**Rating:** 6
**Confidence:** 5
**Correctness:** Experiments have been properly design…

**Review:**

In terms of quality and clarity, the work is excellent. Regarding, originality and significance of this work, it is true what the authors state that is important for the community to have better spatio-temporal datasets, that is way I found the dataset to be of significance, at least to push other researchers on the same direction.

Pros:
The paper is presenting a new multi-modal spatio-temporal dataset that may help the research community towards building richer foundation models and solve complex tasks. I do like the vision-language use case and how relevant is now.

Cons: I miss a face to face comparison with other datasets. The spatial resolution may be valuable for some specific tasks, but it may not be sufficient for medium and high resolution tasks (authors mention it in Section 3 the possibility of including Sentinel2). I miss how the authors iterated through different prompting strategies and what LLM they use to generate them. I only see appendix A available.

**Strengths:**

- Significant contribution towards building better foundation models, covering different modalities.
- Well written paper.

**Additional Feedback:**

What I stated earlier, a mere comparison should be enough.

**Clarity:**

Yes, the paper is well written. I found a small typo in Table 5 caption, second sentence: "UrbanVLP∗ dentoes"

**Documentation:**

It is sufficient.

**Ethics:**

No ethical concern.

**Limitations:**

As I stated, the paper would greatly improved if the authors showed some specific face to face comparison or even using Terra in conjunction with other datasets to demonstrate its value.

**Opportunities For Improvement:**

- A mere comparison with other datasets at the experimentation level.
- Provide more details regarding the experimental setup, such as data examples, how the vision-language was built, etc.

**Relation To Prior Work:**

It seems correct. Perhaps mentioning recent publication such as GeoChat or Skysense.

**Summary And Contributions:**

Authors present a multi-modal spatio-temporal dataset named Terra. The data contains hourly time series from over 6M areas over the world. Authors conduct several experiments aiming at demonstrating dataset's versatility and practicality. In particular, they use Terra for spatio-temporal forecasting (precipitation),  location-based spatial prediction (precipitation, wind speed, temperature) and vision-language based spatial prediction (precipitation, wind speed, temperature).

---

> ### Author Rebuttal · Authors · 2024-08-15
>
> Dear reviewer fRfp:
>
> We are very grateful for the time you spent reviewing this work. We are pleased to hear that you also agree that this work significantly contributes to advancing richer foundational models for the community. Here, we address your concerns and feedback step by step.
>
> + **W1**: I only see appendix A available. Lack of detailed comparisons and introductions.
>
> >   A gentle reminder is that you seem to have overlooked our appendix material, which can extensively address your queries. Following the official requirements, we placed it in the supplementary section, which can be found in the **Supplementary Material**. We apologize for any unnecessary confusion caused by the current document division. Nevertheless, we briefly summarize and respond to your concerns here for your quick understanding:
> >  + **Different prompting strategies**: We compared examples without spatial metadata prompting strategies, which may lead to factual errors, vague descriptions, and incompleteness. As Reviewer 1 pointed out, we also conducted a detailed quality assessment of hallucinations in the existing text.
> >  + **Selection of different LLMs**: Specifically, we selected open-source models including LLaMA3-8B, Vicuna-13b, Gemini-1.5 Pro, and the closed-source model GPT-3.5 for case studies. The results indicate that LLaMA3-8B is the optimal choice in terms of cost and effectiveness.
> >  + **Detailed experimental design**: All detailed training resources, data descriptions, code implementations, and evaluation criteria can be found in Appendix C.
>
>
> + **W2**: Face to face comparisons with other datasets and potential combination opportunities.
>
> > Thank you for suggesting ways to enhance the quality of the paper. Although we have detailed summaries and comparisons of Terra and existing datasets from multiple perspectives in Table 1, we have attempted to provide more specific experimental-level comparisons based on your suggestions:
> >>  Considering the spatio-temporal analysis task, the comparison table with the largest existing dataset, LargeST, is as follows:
> | Datasets | # Unit | Time span | Spatial range | Meta info      |
> |:-:|:-:|:-:|:-:|:-:|
> | LargeST  | 8,600     | 5 year    | California    | One modality   |
> | Our      | 6,480,000 | 45 year   | Global        | Three modality |
>
> >>  Considering the spatial analysis task, the comparison table with the recently proposed UrbanCLIP dataset is as follows:
> | Datasets | Spatial range | # Image | # Text |
> |:-:|:-:|:-:|:-:|
> | UrbanCLIP  | City     | ~1.7w    | ~ 7.6w |
> | Our      | Global | ~150w   | ~26w |
>
> > Additionally, we determine the minimum spatial unit areas through rasterization to ensure the uniform alignment of multimodal data. Therefore, existing external data, such as street view images and POI information, can be easily integrated into our dataset.
>
> + **W3**: Spatial resolution is still insufficient.
>
> >  We understand your concerns. Considering the demand for higher spatial resolution tasks in practical applications, higher resolution is often considered better. We discuss this issue in Appendix D. However, it is important to note that aligning multiple modalities, such as time series, images, and text, on a global scale requires trade-offs. A higher resolution could result in missing data for some modalities, making it currently infeasible, despite our best efforts. Furthermore, due to storage and computational costs, we believe the current dataset is sufficient for most research scenarios.
>
> + **W4**: Typo and mentioning recent publications such as GeoChat or Skysense.
>
> >   We sincerely apologize for this typo error, and we will thoroughly review and refine the manuscript. Additionally, we will discuss the two recent works you mentioned.
>
> Thank you once again for your invaluable suggestions to improve our work!

---

> > ### Comment · Reviewer_fRfp · 2024-08-18
> > **Recommendation for Acceptance**
> >
> > Authors have addressed my comments, but one, and have included further information in their material. Regarding the missing comment addressed, what I meant by comparing their dataset to others was at the experimental level, rather than simply comparing them in terms of figures. Perhaps, I was not clear enough in my review. Despite this, I still recommend the paper for acceptance.

---

> > > ### Author Rebuttal · Authors · 2024-08-28
> > >
> > > Dear Reviewer fRfP,
> > >
> > > Due to the extensive experiments I conducted as per your request, my response has been slightly delayed, and I sincerely apologize for this. I would like to express my heartfelt gratitude for your guidance and recognition. In this round of experiments, we utilized the popular spatio-temporal datasets from the PEMS series. To address your concerns, we conducted a face-to-face comparison from two perspectives. Specifically, we compared the performance differences between the Terra-SA dataset (221 nodes) and Terra-UK dataset (100 nodes) with the most widely used PEMS series datasets (PEMS04: 307 nodes, PEMS08: 180 nodes). The experiments consistently used the past 30 steps to predict the future 30 steps, with the evaluation metrics being the average MAE and RMSE. All experiments were repeated three times to take the average, resulting in a total of 144 experiments. We further analyzed the insights gained from these experiments.
> > >
> > > + **Basic Setting**: In this scenario, we used the full training dataset to train various models.
> > >
> > > > |100% | TimesNet| FEDformer| GWNet|STGCN|
> > > |:-:|:-:|:-:|:-:|:-:|
> > > | **PEMS04** | 31.77 / 52.13| 23.09 / 36.05| 26.52 / 40.54| 20.98 / 33.46|
> > > | **PEMS08** | 26.95 / 44.23| 20.87 / 32.54| 17.79 / 28.53| 17.68 / 28.90|
> > > | **Terra-SA** | 2.08 / 3.46| 2.37 / 3.41| 2.29 / 3.35| 2.28 / 3.41|
> > > | **Terra-UK** | 2.82 / 4.69| 3.38 / 4.66| 3.23 / 4.63| 3.25 / 4.63|
> > >
> > > > **Analysis**: Observing the PEMS series datasets, we found that spatiotemporal graph neural network methods (such as GWNet and STGCN) generally outperform time series neural network forecasting methods (such as TimesNet and FEDformer). However, in our two sub-datasets, spatiotemporal forecasting methods that account for dynamic spatial topological features did not significantly outperform time series forecasting methods that do not consider dynamic topology. Most spatiotemporal and time series forecasting models produced similar results, failing to fully capture spatiotemporal trends. This is intuitive, as precipitation is typically a non-stationary time series with extreme fluctuations [1,2], making it challenging to capture clear patterns. This finding encourages the community to explore ways to improve existing spatiotemporal forecasting models to address such scenarios more effectively. Moreover, integrating more external information to assist in forecasting and enhance interpretability should be considered. As a potential solution, our multimodal dataset Terra offers a promising avenue for further investigation.
> > >
> > > + **Few-Shot Setting**: In this scenario, we trained various models using only the first 5% and 10% of the training dataset, comparing the performance differences between our dataset and the PEMS series datasets.
> > >
> > > > |5%| TimesNet| FEDformer| GWNet| STGCN|
> > > |:-:|:-:|:-:|:-:|:-:|
> > > | **PEMS04**| 63.51 / 92.21| 45.01 / 65.73| 38.12 / 57.80| 31.54 / 47.35|
> > > | **PEMS08**| 51.83 / 76.26| 33.35 / 50.23| 27.65 / 41.74| 28.65 / 42.73|
> > > | **Terra-SA**  | 2.27 / 3.41| 2.71 / 3.66| 2.71 / 3.82| 2.75 / 3.78|
> > > | **Terra-UK**  | 3.03 / 4.57| 4.26 / 5.50| 3.39 / 4.99| 3.92 / 5.39|
> > >
> > > > |10%| TimesNet| FEDformer| GWNet| STGCN|
> > > |:-:|:-:|:-:|:-:|:-:|
> > > | **PEMS04**| 55.68 / 83.14| 41.25 / 61.00| 35.37 / 52.21| 29.85 / 45.46|
> > > | **PEMS08**| 45.34 / 68.56| 31.47 / 47.65| 26.57 / 41.30| 27.95 / 44.10|
> > > | **Terra-SA** | 2.15 / 3.40| 2.58 / 3.51| 2.60 / 3.58| 2.54 / 3.49|
> > > | **Terra-UK** | 2.90 / 4.61| 3.89 / 4.98| 3.41 / 4.76| 3.90 / 5.27|
> > >
> > > > **Analysis**: Compared to pretraining on the full dataset, all methods experienced varying degrees of performance degradation under the few-shot setting. For the PEMS series datasets, existing time series forecasting methods exhibited a more significant performance drop compared to spatiotemporal forecasting methods. However, for our two sub-datasets, the performance decline was similar across all methods, aligning with the phenomenon observed in the full-data setting where existing spatiotemporal and time series forecasting methods yielded comparable results. Additionally, performance degradation was not substantial, indicating the redundancy characteristics of meteorological spatiotemporal data. Given the massive scale of our dataset, this also presents an excellent opportunity to explore research methods in the recently emerging field of data distillation [3,4].
> > >
> > > **In summary**, we believe that our dataset offers rich exploratory potential, providing a new alternative for the analysis of popular spatiotemporal tasks. Moreover, as we also include high-quality multimodal textual and visual supplementary data, we further enhance the dataset's usability. Additionally, due to the gridded partitioning, other existing spatial information datasets from related domains [5,6] can naturally be integrated into our dataset. We hope the above responses address your concerns, and we kindly request you to reconsider your assessment. If you have any further suggestions, we are more than willing to engage in further discussion and make any necessary improvements to the paper.
> > >
> > > Thank you once again for taking the time to improve our paper!
> > >
> > > **Reference**:
> > >
> > > [1] Zheng, Feifei, Seth Westra, and Michael Leonard. "Opposing local precipitation extremes." Nature Climate Change 5.5 (2015): 389-390.
> > >
> > > [2] Cancelliere, Antonino. "Non stationary analysis of extreme events." Water Resources Management 31.10 (2017): 3097-3110.
> > >
> > > [3] Wang, Tongzhou, et al. "Dataset distillation." arXiv preprint arXiv:1811.10959 (2018).
> > >
> > > [4] Lei, Shiye, and Dacheng Tao. "A comprehensive survey of dataset distillation." IEEE Transactions on Pattern Analysis and Machine Intelligence (2023).
> > >
> > > [5] Thomee, Bart, et al. "Yfcc100m: The new data in multimedia research." Communications of the ACM 59.2 (2016): 64-73.
> > >
> > > [6] Mooney, Peter, and Marco Minghini. "A review of OpenStreetMap data." Mapping and the citizen sensor (2017): 37-59.

---

> > > > ### Comment · Reviewer_fRfp · 2024-08-29
> > > > **Great follow up**
> > > >
> > > > Great work! You definitely have my recommendation for acceptance.

---

> > > > > ### Author Response · Authors · 2024-08-29
> > > > > **Sincere Gratitude from the Authors**
> > > > >
> > > > > Dear Reviewer fRfp:
> > > > >
> > > > > We are genuinely pleased that our responses have effectively addressed your concerns. We sincerely appreciate the time you have taken to review our paper and provide us with such detailed and invaluable feedback.

---

### Official Review · Reviewer_A8vf · 2024-07-24

**Rating:** 7
**Confidence:** 5
**Correctness:** Yes
**Clarity:** Yes

**Review:**

Pros:

- This extensive temporal and spatial coverage of Terra ensures robustness and generalizability in research.

- The dataset supports fine granularity with up to 3-hourly time intervals and 0.1° spatial resolution, enabling detailed analysis and applications.

- Incorporating geo-images and textual information benefits traditional time series analysis.

- The paper is well-organized, with clear explanations of the dataset's components, methodologies for data collection, and potential applications.

- The authors provide detailed use cases and experimental setups, providing practical instructions of how the dataset can be utilized.

Cons:

- The multimodal datasets are somewhat too complicated. The vast amount of data and its multimodal nature might pose a challenge for researchers in terms of data processing and integration.

**Strengths:**

See review

**Additional Feedback:**

None

**Documentation:**

Yes

**Opportunities For Improvement:**

- The complexity of processing and integrating such a vast and diverse dataset can be challenging.

- Despite the extensive temporal coverage, there might be periods with missing or sparse data due to limitations in data collection technologies over the 45-year span.

**Relation To Prior Work:**

Yes

**Summary And Contributions:**

The paper introduces  a comprehensive, multimodal spatio-temporal dataset that spans the entire Earth. The dataset addresses the limitations of existing public datasets in terms of spatial scale, temporal coverage, and multimodality.  The dataset supports research in spatio-temporal data mining and advance towards general intelligence in this domain. The source code and data are available.

---

> ### Author Rebuttal · Authors · 2024-08-15
>
> Dear reviewer A8vf:
>
> We sincerely appreciate your time reviewing our work, and we are delighted that you recognize the richness and practicality of our dataset. Thank you for pointing out opportunities for further improvement. Here, we strive to address some of your concerns.
>
> + **W1**: Multimodal datasets are a bit too complex.
>
> >  We acknowledge that excessively large and complex datasets have both advantages and disadvantages, which may pose significant challenges for data processing and analysis. Therefore, we provide multiple versions of dataset variants to accommodate different research requirements. In addition, we determine the minimum spatial unit areas for different versions through rasterization to ensure the uniform alignment of multimodal data.
>
> + **W2**: Data processing and integration are complex.
>
> >  For researchers with limited computing resources, we recommend using the coarsest-grained dataset (i.e., with a spatial resolution of 1° and a temporal resolution of one month). This allows researchers to easily use data at this scale for rapid innovation in certain research methodologies. Additionally, we organize the current data into commonly used and accessible formats for each modality in an appropriate manner. Finally, we believe that if the community shows interest in this work, it will collaboratively advance and maintain the project, making it more popular and user-friendly.
>
> + **W3**: Missing or sparse data due to 45-year span.
>
> >  We acknowledge that such a long time span inevitably leads to some missing and sparse data due to sensor technology. However, the data from Terra (particularly the time-series modality) is high-quality, collected and analyzed through multiple sensor sources. Here, we further analyze the missing data for all years and find that there is no missing data.
>
> Thank you once again for your invaluable suggestions to improve our work!

---

> > ### Comment · Reviewer_A8vf · 2024-08-19
> >
> > Thanks for the authors' response. The authors should provide more details of using this multimodal data for different purposes in the camera-ready version. I think some examples will be benefit. Overall, I recommend to accept this paper with necessary revision.

---

> > > ### Author Response · Authors · 2024-08-28
> > > **Sincere Gratitude from the Authors**
> > >
> > > Dear Reviewer A8vf,
> > >
> > > I sincerely apologize for the slight delay in responding due to dealing with a large number of experiments requested by another reviewer. I would like to express my gratitude for your guidance and feedback. Specifically, we have provided six combinations of mult-imodal spatio-temporal tasks and examples in Table 2 of the manuscript. In the revised version, we will include more formal explanations and definitions for each task. Additionally, we have noted the recent emergence of surveys[1, 2] and studies[3~6] that combine foundational models (text and vision) with spatiotemporal sequences, and we will incorporate these into the final revised version.
> > >
> > > Once again, thank you very much for your valuable guidance!
> > >
> > > **Reference**:
> > >
> > > [1] Ye, Jiexia, et al. "A Survey of Time Series Foundation Models: Generalizing Time Series Representation with Large Language Mode." arXiv preprint arXiv:2405.02358 (2024).
> > >
> > > [2] Jin, Ming, et al. "Large models for time series and spatio-temporal data: A survey and outlook." arXiv preprint arXiv:2310.10196 (2023).
> > >
> > > [3] Li, Zhe, et al. "Orca: Ocean Significant Wave Height Estimation with Spatio-temporally Aware Large Language Models." arXiv preprint arXiv:2407.20053 (2024).
> > >
> > > [4] Xu, Zhijian, et al. "Beyond Trend and Periodicity: Guiding Time Series Forecasting with Textual Cues." arXiv preprint arXiv:2405.13522 (2024).
> > >
> > > [5] Yoon, Hyungjun, et al. "By My Eyes: Grounding Multimodal Large Language Models with Sensor Data via Visual Prompting." arXiv preprint arXiv:2407.10385 (2024).
> > >
> > > [6] Yang, Luoxiao, et al. "ViTime: A Visual Intelligence-Based Foundation Model for Time Series Forecasting." arXiv preprint arXiv:2407.07311 (2024).

---

### Official Review · Reviewer_EWMy · 2024-07-24

**Rating:** 7
**Confidence:** 4
**Clarity:** The paper is well written.

**Review:**

Strengths
---

* This dataset is an ambitious effort, and I applaud the authors for taking this on.
* The multimodal nature of the dataset, which includes timeseries + image and timeseries + text samples, enhances the originality and utility of this work. There is a severe lack of multimodal timeseries + X datasets, so it is nice to see some work in this space.
* The paper is well-written and comprehensive: The data collection process, hosting, and licensing are detailed, and various intended use-cases are described.
* The diversity of use-cases for the single dataset are impressive, e.g., from spatial-temporal timeseries forecasting (4.1) to vision-language based spatial variable prediction (4.2.2).

Weaknesses and opportunities for improvement
---

* However, I am concerned that this dataset is a little *too* ambitious. It is worrisome that only a small subset of the data has been released publicly on HuggingFace so far, and a vague promise is made to release the rest. Moreover, the paper describes "enormous time and monetary cost" related to supporting text and images at a spatial resolution of 0.1'. There are a lot of applications that need spatial resolutions $\leq$ 0.1'. Already, a temporal resolution of 3 hours is limiting in that many applications (for example, power systems) depend on 1 hour temporal resolution.

**Strengths:**

In addition to the strengths discussed above, I believe this work would be of broad interest to the multimodal ML community and the AI for Climate Change/Social Good communities.

**Additional Feedback:**

N/A

**Correctness:**

The claims made are mostly correct. The text generation approach with LLMs is promising; however, it is unclear to what extent, even with metadata, the LLM-generated text is reliable. One suggestion to quantify is to a) extract a text embedding from the generated text, e.g., using the LLM itself or BERT, then b) train a multi-class classifier or regressor on top of this embedding to predict each metadata attribute present in the txt metadata file. The accuracy of the classifier can act as a rough proxy for the hallucination-rate per attribute (how often the LLM leaves out an attribute or changes it erroneously). See [1] for an example of this.


References:

[1] Emami, Patrick, et al. "SysCaps: Language Interfaces for Simulation Surrogates of Complex Systems." arXiv preprint arXiv:2405.19653 (2024).

**Documentation:**

The dataset is well documented. A URL is provided but only a subset of the dataset is currently available.

**Limitations:**

The authors are upfront about current limitations of the dataset.

* L207-208 mention a commitment to continually invest time and money to add more "higher resolution" (the paper says "lower resolution", but I think "higher" was the intent?) text and image modalities. As mentioned above, I have a concern that the overall dataset may be too ambitious, and supporting/maintaining it will be highly difficult.
* In the appendix, the authors also acknowledge the issue of timeliness. A foundational dataset for the Earth will necessarily need to be continually updated, and models trained on these datasets will constantly be going "out of date". It is good that the work mentions this.

**Opportunities For Improvement:**

Please release all of the data publicly as soon as possible, or provide a good reason why this cannot be done now. The final review score will depend on the accessibility of this data, as it is the primary contribution.

**Relation To Prior Work:**

Yes

**Summary And Contributions:**

This paper introduces a large-scale spatiotemporal dataset, Terra, for the Earth. It includes spatially localized timeseries for various atmospheric variables, satellite imagery and geo images, and geospatial text descriptions. The dataset has the potential to be used for spatiotemporal data mining research. A subset of the dataset has been publicly released on HuggingFace.

---

> ### Author Rebuttal · Authors · 2024-08-15
>
> Dear reviewer EWNY:
>
> We sincerely appreciate your time and effort in providing insightful feedback on our paper. It is a great honor to have your approval of our hard work. We have carefully considered each of your comments and addressed them one by one.
>
> + **W1**: The dataset is a bit too ambitious, and currently only a part of it is publicly available on hugging face.
>
> > We sincerely appreciate your recognition of the substantial effort and complexity involved in creating such a comprehensive dataset. Firstly, we have fully released all the datasets used in the experimental section, which can be found in the current anonymous repository. During the review process, we only uploaded a portion of the dataset for review purposes, with a commitment to formally release the entire dataset upon notification. However, we are pleased that all reviewers unanimously acknowledged and recognized the importance of our dataset, and therefore, we intend to begin uploading the data in batches from now on. Additionally, all data have been processed and are ready on our server. Below, we provide the size statistics of different modalities at the finest granularity:
> |||Time|series|Modality||||$\|$Image|Modality|$\|$Text|Modality|
> |:---:|:---:|:---:|:---:|:---:|:---:|:---:|:---:|:---:|:---:|:---:|:---:|
> |pres|prec|wind|temp|swd|lwd|speh|relh|  $\|$  geo | satellite |  $\|$ meta|llm|
> |1.1T|282G|532G|485G|609G|921G|355G|731G|$\|$45g|108g| $\|$  20mb| 120mb|
>
> > It can be observed that our current dataset, even at its finest granularity, amounts to approximately 5TB, which is nearly 26 times larger than ImageNet (~200GB) by comparison. Therefore, considering the instability and speed of the network for uploading such a large dataset, we estimate that this process will take approximately two-three months. **It is worth noting that we have carefully reviewed the SUBMISSION INSTRUCTIONS, which clearly state, "when the data can only be released at a later date, this can be added afterward (up to a year after the submission deadline)."** Thus, we believe that our upload strategy, which balances resources and time, is a completely acceptable choice and should alleviate any unnecessary concerns you may have.
>
> + **W2**: Some applications still need ≤ 0.1° or <= 3 hour.
>
> > Although we have made our best effort to collect the largest, highest-resolution, and most diverse data sources available, as you mentioned, the benefits of higher spatiotemporal resolution in practical applications are still worth considering. That being said, achieving such a resolution in the context of our current spatiotemporal dataset is nearly impossible. Please note that our dataset covers a global scale and spans 45 years; collecting data with even higher spatiotemporal resolution on this scale would incur prohibitively high storage and computational costs (as shown in the previous table). Therefore, we believe that the current version is already sufficient for researchers to conduct preliminary and in-depth studies.
>
> + **W3**: Supporting / Maintaining it will be highly difficult. (the issue of timeliness).
>
> > We acknowledge that an excessively large and complex dataset has both advantages and disadvantages, as it may pose significant challenges for data processing and analysis. To address this, we have provided multiple versions of the dataset in the paper, each tailored to meet different research requirements. Additionally, we have used a rasterization approach to define the smallest spatial unit regions across different versions, ensuring uniform alignment of the data across modalities. As a result, existing external data, such as street view images and POI information, can be easily integrated into our dataset. Moreover, we would like to point out that, considering most current methods are tested on datasets with limited temporal coverage, our dataset appears to be sufficiently large to support research for a considerable period. Of course, regular maintenance is necessary, but we only need to keep appending data for new years, so the maintenance cost is acceptable. Finally, we believe that if the community shows interest in this work, it will lead to collaborative efforts to advance and maintain this project, making it more popular and user-friendly.
>
> + **W4**: Using experiments to solve the problem of hallucinations.
>
> > We greatly appreciate the authors for providing a method to effectively detect hallucination risks. However, this method is very recent (5/30/2024) and appears to still be under review, so no reliable official implementation is available yet. Despite this, we have made every effort over the past few days to understand the method and have attempted to reproduce a version for application to hallucination detection in our textual data. Specifically, based on your suggestion, we referred to this paper to calculate the hallucination rate of the generated text. Below, we present the NRMSE and Acc metrics for several regions of text used in the experiments of our paper:
> ||NRMSE|Accuracy|
> |:-:|:-:|:-:|
> | USA | 0.0519 | 0.764 |
> | AUS | 0.056  | 0.724 |
> | SA  | 0.0705 | 0.696 |
>
> >  Due to the diverse nature of our metadata, we simultaneously train a regressor and a classifier. For the regression task, we predict the elevation values and latitude/longitude coordinates of the current region. For the classification task, we identify the predominant land vegetation type in the area. Additionally, we chose BERT as the backbone encoder. The dataset is split according to a 7:1:2 ratio. The experimental results demonstrate that our spatial prompt engineering method achieves a retention rate and success rate of approximately 70%, effectively mitigating the hallucination issue to some extent. We plan to explore more quantitative experiments in future versions. We sincerely appreciate your valuable feedback, which has further ensured the quality of our work.
>
> Thank you once again for your invaluable suggestions to improve our work!

---

> > ### Comment · Reviewer_EWMy · 2024-08-28
> >
> > I thank the authors for taking my feedback and comments into consideration. I am satisfied with their responses and have raised my score to reflect a recommendation to accept the paper.

---

> > > ### Author Response · Authors · 2024-08-28
> > > **Sincere Gratitude from the Authors**
> > >
> > > Dear reviewer EWNY:
> > >
> > > We are truly delighted that our responses have effectively addressed your concerns, and we would like to express our sincerest gratitude once again for taking the time to review our paper and provide us with such detailed and invaluable comments.

---

### Decision · Program_Chairs · 2024-09-26

**Decision:**

Accept (Poster)

**Comment:**

The reviewers see several strengths in the paper:
- s1. huge multimodal datasets comprising time series, images and text.
- s2. comprehensive description of data set curation.
- s3. several different tasks / use cases discussed.

But they also discussed several major issues:
- w1. data only partially released yet.
- w2. size of the data may hinder its uptake by the research community.
- w3. the temporal and/or spatial resolution may be too low for some applications.
- w4. no detailed comparison with other datasets.

During the rebuttal the authors addressed all raised major issues:
they answer w1 "only partial data yet" by promising to upload the data
upon acceptance, w2 "size too large for uptake by others" by providing
subsets, w4 "no comparison with other datasets" by a comparison
with LargeST. They discuss w3 "too low temporal/spatial resolution"
as a principled trade off between being too large to handle and too
low resolution for some applications.

Overall I therefore recommend to accept the paper.